# Mesenchymal Stem Cells in the Adult Human Liver: Hype or Hope?

**DOI:** 10.3390/cells8101127

**Published:** 2019-09-22

**Authors:** Irina V. Kholodenko, Leonid K. Kurbatov, Roman V. Kholodenko, Garik V. Manukyan, Konstantin N. Yarygin

**Affiliations:** 1Orekhovich Institute of Biomedical Chemistry, 10, Pogodinskaya St., 119121 Moscow, Russia; leonid15@mail.ru (L.K.K.); kyarygin@yandex.ru (K.N.Y.); 2Shemyakin-Ovchinnikov Institute of Bioorganic Chemistry, Russian Academy of Sciences, 16/10, Miklukho- Maklaya St., 117997 Moscow, Russia; khol@mail.ru; 3Petrovsky Russian Research Center of Surgery, 2, Abricosovskiy Pereulok, 119991 Moscow, Russia; drmanukyan@mail.ru

**Keywords:** human liver mesenchymal stem cells, differentiation potential, cell phenotype, cirrhosis, fibrosis, immunomodulation, cell therapy

## Abstract

Chronic liver diseases constitute a significant economic, social, and biomedical burden. Among commonly adopted approaches, only organ transplantation can radically help patients with end-stage liver pathologies. Cell therapy with hepatocytes as a treatment for chronic liver disease has demonstrated promising results. However, quality human hepatocytes are in short supply. Stem/progenitor cells capable of differentiating into functionally active hepatocytes provide an attractive alternative approach to cell therapy for liver diseases, as well as to liver-tissue engineering, drug screening, and basic research. The application of methods generally used to isolate mesenchymal stem cells (MSCs) and maintain them in culture to human liver tissue provides cells, designated here as liver MSCs. They have much in common with MSCs from other tissues, but differ in two aspects—expression of a range of hepatocyte-specific genes and, possibly, inherent commitment to hepatogenic differentiation. The aim of this review is to analyze data regarding liver MSCs, probably another type of liver stem/progenitor cells different from hepatic stellate cells or so-called hepatic progenitor cells. The review presents an analysis of the phenotypic characteristics of liver MSCs, their differentiation and therapeutic potential, methods for isolating these cells from human liver, and discusses issues of their origin and heterogeneity. Human liver MSCs are a fascinating object of fundamental research with a potential for important practical applications.

## 1. Introduction

About half a century ago, Friedenstein et al. [1,2] isolated and characterized bone marrow fibroblast-like colony-forming cells of mesenchymal origin and called them bone marrow stromal cells (BMSCs). BMSCs were shown to enhance the viability and support normal activity of hematopoietic stem cells and could be induced to differentiate into osteoblasts in vitro and heterotopically in vivo. Later, their ability to undergo differentiation into other cells of the mesodermal lineage including chondrocytes, adipocytes and myocytes was reported [3,4], thus supporting the concept of mesenchymal stem cells representing a pool of connective tissue progenitors, first suggested by Caplan [5]. Cells with similar characteristics and differentiation potential were promptly isolated from many other tissues [6] and also named mesenchymal stem cells (MSCs). Eventually, MSCs proved to be able to differentiate into ecto- and endodermal derivatives [7,8]. Gradually it became clear that MSC cultures are not homogenous and comprise cells with different physiological functions and varying differentiation capacity [9]. It is still widely accepted that despite certain minor dissimilarities, MSCs residing in different tissues have identical or very close phenotype and characteristics. However, accumulating evidence suggests more pronounced MSC tissue specificity than usually believed, though, of course, all MSCs share a set of common features determining that they are MSCs and not another cell type. Using more stringent assays several research groups provided clear evidence that MSCs from different tissues have intrinsically diverse transcriptomic signatures resulting in fundamental differences of properties including differentiation capacities [10,11,12]. In-depth studies of tissue-specific MSCs have obvious practical applications, because MSCs are the basis of many cell therapies under development.

Chronic liver diseases constitute a significant economic, social, and biomedical problem worldwide. By some estimates, more than 800 million people around the world suffer from chronic liver diseases and about 2 million of them die every year. Cirrhosis is the end-stage of chronic liver disease. In most patients, cirrhosis stays asymptomatic for a long time, which leads to late diagnosis of the disease. Decompensated cirrhosis resulting in about 1 million deaths annually is the 14th most common global cause of death in adults [13]. The major causes of cirrhosis are chronic liver damage, such as infection with hepatitis C and B viruses (HCV and HBV), alcoholic liver disease, and non-alcoholic fatty liver disease (NAFLD). Less common liver pathologies that can lead to the development of the end-stage liver disease include autoimmune hepatitis, primary biliary cirrhosis, primary sclerosing cholangitis, and hemochromatosis-associated cirrhosis [14]. Conservative treatment of end-stage liver disease is symptomatic and only organ transplantation can considerably help patients with cirrhosis. However, due to social and economic problems, only a small number of patients receive transplantation. In addition, a long and expensive immunosuppressive therapy is required after liver transplantation, and the risk of transplant rejection is always there. For these reasons, while it remains the only effective way to treat end-stage liver disease, liver transplantation does not meet current clinical and medical requirements in this area. 

Cell therapy as a treatment for chronic liver disease yielded quite satisfactory preclinical and clinical results. Transplantation of hepatocytes, first performed in animals with metabolic disease as early as in 1976, is the oldest method of cell therapy of liver pathology [15]. To date, human hepatocyte transplantation has been used to treat different liver diseases including factor VII deficiency, infantile Refsum disease, severe infantile oxalosis, urea cycle disorders, phenylketonuria, glycogen storage disease type 1, and acute liver failure [16]. Even though the clinical safety of hepatocyte transplantation is proven, the effectiveness of this treatment is low and transitory. In addition, this method has limitations associated with the cellular material itself. First, the access to donor livers suitable for obtaining large numbers of high-quality hepatocytes is limited and represents a rather serious problem. Second, the problem of cultivating, expanding, passaging, and cryopreservation of hepatocytes, as well as maintaining their viability and functionality is still not resolved. Third, hepatocyte engraftment in the recipient liver is very low; in mice it accounts for 0.1 to 0.3% of host liver mass after infusion of donor cells in numbers equivalent to 3–5% of the total number of recipient’s liver cells [17]. 

An alternative liver disease cell therapy approach is the use of stem/progenitor cells capable of differentiating into functionally active hepatocyte-like cells. In particular, good results in the treatment of liver pathologies were obtained using MSCs obtained from various tissue sources. MSCs have been shown to be effective in treating liver failure in both animal models and clinical trials [18,19]. There is an assumption that resident MSCs present in the same tissue to which cellular therapy is directed are the most effective in treating various diseases. This assumption is mainly based on the fact that resident MSCs demonstrate certain tissue-specific commitment and are likely to be able to differentiate into the desired cell type more effectively and more efficiently engraft into the corresponding recipient tissue [12]. Therefore, in-depth studies of resident liver MSCs are of prime importance for the development of cell-based therapies.

The aim of this review is to analyze the results regarding the study of liver MSCs that probably are one more kind of liver stem/progenitor cells different from hepatic stellate cells and so-called hepatic progenitor cells (human analog of rodent oval cells). Just several research groups in the world focus their studies on this type of cells; they distinguish the cells in question as an independent population of liver stem/progenitor cells that possesses the properties of classical MSCs isolated from extrahepatic sources and at the same time expresses a range of liver-specific genes. These liver MSCs effectively differentiate into hepatocyte-like cells in vitro and in vivo and promote liver regeneration in animal models of liver failure. A Phase I/II clinical trial was conducted using liver MSCs to treat pediatric patients with metabolic disorders. Since no generally accepted designation for liver MSCs exists, different authors refer to these cells in different ways, for example, adult-derived human liver stem/progenitor cells (ADHLSCs) [20], human liver stem cells (HLSCs) [21], or hepatic mesenchymal stem cells [22], in this review, for convenience, we will use the simplest name “liver mesenchymal stem cells” (liver MSCs). The review presents a detailed analysis of the phenotypic characteristics of liver MSCs, their differentiation and therapeutic potential, methods of isolation these cells from the human liver, and discusses issues relating to the origin of these cells.

## 2. Methods of Isolation of Human Liver MSCs

In most of the published studies, human liver MSCs were obtained from the parenchymal fraction of the liver. Isolation protocols used by different authors vary to some extent, although perfusion and tissue treatment with collagenases are used most often. As a rule, subsequent cultivation and expansion of liver MSCs is carried out under standard conditions employed for cultivation of MSCs isolated from other sources, i.e., in Dulbecco’s modified Eagle’s medium (DMEM) with the addition of 10–20% FBS. 

Najimi et al. [20] isolated liver MSCs from the left liver segment of the cadavers of people who died at different ages from injuries and displayed no serious pathological conditions before death. The liver segments were sequentially perfused with ethylene glycol bis(beta-aminoethyl ether)-N,N,N′,N′-tetraacetic acid (EGTA) solution and a digestion enzyme solution containing collagenase P. The resulting suspension consisted of 95% mature hepatocytes. The cells were seeded on plates coated with type I collagen in hepatocyte culture medium (Williams’ E medium supplemented with 10% fetal calf serum (FCS), EGF, insulin, dexamethasone, and penicillin/streptomycin). Following 24 h, non-adherent cells were removed, and culturing of adherent cells in DMEM supplemented with 10% FCS and 1% penicillin/streptomycin was continued, resulting in complete elimination of hepatocytes and expansion of mesenchymal cells.

Herrera et al. isolated human liver MSCs from fresh surgical specimens of patients undergoing hepatectomy by previously obtained a suspension of hepatocytes [21], or, alternatively, from cryopreserved hepatocytes [23]. The original hepatocyte suspension was seeded in Williams Medium E supplemented with glutamine and 5% FCS. Non-adherent cells were removed after 2–3 h, and hepatocyte serum-free medium was added to the culture plates. The hepatocytes died in 2 weeks, and the medium was replaced with α-minimum essential medium/endothelial cell basal medium-1 (α-MEM/EBM) (3:1) supplemented with l-glutamine, HEPES, penicillin/streptomycin, FCS (10%), and horse serum (10%). Individual adherent cells were identified on the plates within the next 3 weeks, followed by formation of colonies which were further subcultured, each clone separately.

Pan et al. [22] obtained liver MSCs from liver graft preservation fluids (perfusates) initially isolating mononuclear cells by density gradient centrifugation using Ficoll Paque Plus. Cells were cultivated in DMEM supplemented with 10% fetal bovine serum and penicillin/streptomycin, i.e., in standard conditions for MSC cultivation. Perfusates collected during liver transplant processing contained high numbers of mononuclear cells that come out from the liver including lymphocytes, natural killer cells, antigen-presenting cells [24,25], and hematopoietic stem cells [26]. In addition, the perfusates contained minute numbers of cells that are double-positive for surface mesenchymal markers CD90 and CD105 (0.09% ± 0.07%) and for CD90 and CD166 (0.02% ± 0.02%) [22]. The authors also obtained mesenchymal cells with similar characteristics from explanted livers taken out from different patients with end-stage liver disease by tissue dissociation and cultivation of unfractionated cell suspensions under standard MSC culture conditions. After 4–10 days, clusters of cells with fibroblast-like morphology were observed in most of these cultures. As with MSCs from perfusates, these cells were highly positive for CD90, CD105, and CD166, and negative for CD34, CD45, and HLA-DR. Liver MSC cultures can also be derived from disease-free liver graft tissue obtained from post-mortem donors [22]. 

Lee et al. [27,28] isolated a cell population with mesenchymal cell characteristics from the non-neoplastic part of the liver of patients who underwent resection of hepatic hemangioendothelioma, or, alternatively, from a whole organ that was not suitable for transplantation. After liver perfusion, the authors performed three centrifugations at 50× *g* to remove hepatocytes; thus, the non-parenchymal fraction was used to obtain mesenchymal cells in these studies. Subsequent cultivation of cells was carried out under standard conditions used for the cultivation of MSCs, namely in DMEM supplemented with 10% FBS.

We [29], as well as other authors [30,31], obtained liver MSCs from liver biopsy material by tissue disintegration and subsequent processing with collagenase. The resulting cell suspension, without any additional manipulations, was placed in uncoated culture flasks and incubated in the medium for the cultivation of MSCs (DMEM + 10–20% FBS).

Thus, human liver MSCs have been obtained from several sources, including parenchymal and non-parenchymal portions of liver tissue and mononuclear cells released from donor liver into the graft preservation fluids. As with other MSCs cultures, stable human liver MSC cultures can be established and maintained in DMEM supplemented with FBS.

## 3. Morphology and Phenotype of Human Liver MSCs

Morphologically, cultured liver MSCs do not differ from human MSCs isolated from other tissues, i.e., they have a spindle-shaped fibroblast-like morphology (Figure 1A). Their morphology may change after prolonged (several months) passaging: in 2D cultures they spread to cover a larger area compared to the same cells during early passaging (Figure 1B).

### 3.1. Mesenchymal Stem/Stromal Cell Markers

The phenotype of liver MSCs largely coincides with the phenotype of MSCs isolated from other tissue sources. These cells express mesenchymal markers such as CD29, CD44, CD73, CD90, HLA-Class I, etc. [32]. Still, the expression levels of these markers vary in reports from different authors. For example, Najimi et al. [20] found that 99% of liver MSCs were positive for CD90, 92% of the cells were positive for CD73, 88% were positive for CD29, 92% for CD44, and 76% for HLA-Class I. In our work, we showed by flow cytometry that only about 30% of liver MSCs isolated from the liver of patients with cirrhosis and fibrosis expressed CD90 and CD44 [33]. Moreover, a gene expression microarray confirmed that the expression level of CD90 (*THY1*) was low, while the expression level of *CD44* was very high (our unpublished data; see Table 1). Beltrami et al. [34] demonstrated that most liver MSCs (more than 90% of cells in the population), as well as MSCs isolated from the heart and bone marrow, expressed CD13, CD49b, and CD90 at a high level (high MFI values according to flow cytometry), while the main part of cells in the population (80–90%) expressed low levels of CD73, CD44, HLA-ABC, CD29, CD105, and CD49a (low MFI values according to flow cytometry). Flow cytometry analysis of liver MSCs isolated from the mononuclear fraction of the perfusate collected during liver transplantation and maintained in culture showed a profile of surface markers typical for mesenchymal stem cells: CD90^+^ (59% ± 18%), CD105^+^ (55% ± 14%), and CD166^+^ (44% ± 16%) between passages 4 and 9 also [22].

The Mesenchymal and Tissue Stem Cell Committee of the International Society for Cellular Therapy proposed minimal criteria to define human MSCs [35]. Among others, three markers indicating that cells in a population are mesenchymal stromal cells, namely CD44, CD90, and CD105, were suggested. While the expression of the first two markers (CD44 and CD90) by liver MSCs has been consistently shown (though to varying degrees), data regarding the expression of CD105 differ greatly among the authors. Indeed, Najimi et al. in their earlier work [20] showed the absence of CD105 expression on the surface of liver MSCs. This result is consistent with flow cytometry data by Beltrami et al. [34], although, in the later work, the up-regulation of the *ENG* (endoglin) gene transcription at the mRNA level was demonstrated. We also showed that liver MSCs do not express CD105 on their surface, although its expression at the transcriptome level is enhanced. Herrera et al. [21] demonstrated by flow cytometry that no more than 20% of liver MSCs were positively stained with anti-CD105 antibodies, and the expression of this marker varied greatly from clone to clone. Kellner et al. [30] isolated mesenchymal stromal cells from bone marrow, heart, adipose tissue, and liver and showed that 100% of the cells in these four populations expressed CD105. Comprehensive screening of surface markers of liver MSCs also showed that more than 90% of the cells express endoglin (CD105) [36], which contradicts the data reported by the same team several years earlier [20].

Liver MSCs also express intracellular mesenchymal markers such as vimentin and α-SMA (α-smooth muscle actin) [20], fibronectin [34,37], as well as the marker of resident stem/progenitor cells nestin [28].

### 3.2. Hematopoietic/Endothelial Cell Markers

As with MSCs isolated from other sources, liver MSCs do not express markers of hematopoietic and endothelial cells, such as CD11b, CD14, CD19, CD31, CD34, CD45, CD79β, CD117, CD133, CD144, and HLA-DR [20,23,27,33,36]. The lack of expression of CD34, CD45, and CD117 on the surface of liver MSCs suggests that these cells are not so-called hepatic progenitor cells (the human counterpart to rodent oval cells) which are bipotent resident hepatic cells that can differentiate into hepatocytes and cholangiocytes [28,38].

### 3.3. Pluripotency Markers

Regarding the expression of pluripotent stem cell markers, the results obtained by different authors are very contradictory. A large-scale screening of the surface markers of liver MSCs showed that these cells do not express such pluripotent stem cell markers as SSEA-3 and -4, Tra1-60, and Tra1-81 [36]. The only putative pluripotent cell marker found to be highly expressed was CD13 (membrane alanyl aminopeptidase). However, though once CD13 was assumed to be a pluripotency marker [39], later it was detected on many cell types, not just on adult stem/progenitor cells [40], but also on various types of mature fully differentiated cells, such as endothelial cells [41], dendritic cells [42], and others. Moreover, CD13 has been listed as a major marker of mesenchymal stem/stromal cells, but not embryonic stem cells or induced pluripotent stem cells [43,44]. It was demonstrated high CD13 expression on the surface of liver MSCs, along with MSCs derived from bone marrow and umbilical cord. The function of CD13 on the surface of MSCs is to regulate the activation of FAK (focal adhesion kinase) and, therefore, to improve cell adhesion to the extracellular matrix and enhance cell migration; CD13 is also an important angiogenic regulator [45,46]. On the other hand, Herrera et al. [21] revealed that liver MSCs express embryonic stem cell markers NANOG, Oct-4, SOX2, and SSEA4. Beltrami et al. [34] also showed that adult human liver contains mesenchymal stem cells similar to bone marrow and cardiac MSCs, which show a low level of tissue commitment, express specific pluripotency markers on the mRNA level, such as Oct-4, REX1, and NANOG, and exhibit telomerase activity. Our gene expression microarray data show that liver MSCs isolated from liver biopsies of patients with cirrhosis and fibrosis do not express *NANOG*, REX-1 (*ZFP42*), and *SOX2*. The expression of Oct3/4 (*POU5F1*) varied in different patients, and was in general low (see Table 1).

### 3.4. Integrins and Adhesion Molecules

Since MSCs of different origin possess high migration ability and can undergo homing in various tissues and organs [47,48], expression of adhesion molecules on the cell surface represents an important phenotypic characteristic of these cells. To reach the parenchyma of the organ after systemic administration, MSCs should behave like leukocytes do during inflammation. Specifically, the cells first need to slow down inside the vascular endothelium by means of selectin ligands, followed by firm adhesion to endothelial proteins, such as ICAM and VCAM-1, through integrin heterodimers, VLA-1 (consists of alpha chain α1 (CD49a) and beta chain β1 (CD29)) and VLA-4 (consists of alpha chain α4 (CD49d) and beta chain β1 (CD29)), respectively. Like most MSCs, liver MSCs do not express CD162 (PSGL-1), the high-affinity receptor for P-selectin, and also do not express sialyl-Lewis X (CD15s or SSEA-1), which is required for E-selectin binding [36]. Thus, as with other mesenchymal cells, liver MSCs probably employ an alternative mechanism of adhesion and rolling along the vascular endothelium. In particular, it is supposed that liver MSCs can use the hyaluronic acid receptor CD44 highly expressed on the cell surface [49]. A similar adhesion/rolling mechanism has been shown for neutrophils—their adhesion to the sinusoidal endothelium in the event of an inflammatory process in the liver depends on the direct binding of CD44 on the surface of neutrophils to hyaluronan expressed by endothelial cells [50]. As with MSCs obtained from other sources, liver MSCs manifest a high expression of the integrin subunit beta 1 (or CD29) [6,20,21,27]; however, reports on the expression of integrin α-chains, which are part of the VLA-1 and VLA-4 heterodimers responsible for the interaction with ICAM and VCAM-1 on the vascular endothelium, vary. Dollet et al. [36] did not detect VLA-1 expression by liver MSCs at either the protein or the mRNA level; VLA-4 was expressed on the cell surface at a low level, although constitutive high expression was manifested at the mRNA level. Our gene expression microarray data show that liver MSCs obtained from the liver of patients with fibrosis and cirrhosis express medium levels of VLA-1 (CD49a; *ITGA1*) and low levels of VLA-4 (CD49d; *ITGA4*). Beltrami et al. [34] showed by flow cytometry that approx. 70–80% of liver MSCs express CD49a at a low level. Weak positivity for CD49a in MSCs isolated from the liver was also shown [36]. Most studies showed that the expression levels of integrins on liver MSCs responsible for interaction with the extracellular matrix are rather high. Specifically, liver MSCs express VLA-2 (CD49b; integrin subunit alpha 2; *ITGA2*) that binds collagen, VLA-3 (CD49c; integrin subunit alpha 3; *ITGA3*) that binds laminin, and VLA-5 (CD49e; integrin subunit alpha 5; *ITGA5*) that binds fibronectin [28,34,36]. According to our gene expression microarray results (see Table 1), in addition to the above-mentioned integrins, liver MSCs also express integrin subunit alpha 11 (*ITGA11*) which interacts with collagen, integrin alpha-V beta 3 (CD51; *ITGAV*)—interacts with extracellular matrix proteins (e.g., vitronectin, fibronectin, fibrinogen, and thrombospondin), integrin subunit alpha 7 (*ITGA7*) which is a laminin receptor, and integrin subunit alpha E (CD103; *ITGAE*) which interacts with E-cadherin and several other integrins [51,52].

In the tissues, cell-cell adhesion is mediated through the interaction of various cell adhesion molecules (CAMs), including cadherins, integrins, selectins, and immunoglobulin-like CAMs. In addition to adhesion, these molecules also regulate a wide range of cellular processes, such as proliferation, differentiation, apoptosis, and self-renewal of pluripotent stem cells, including embryonic stem cells and induced pluripotent stem cells [53,54]. Cadherins, transmembrane glycoproteins that mediate Ca2^+^-dependent homophilic interactions between the cells, represent one of the classes of CAMs. Different types of cadherins are expressed by different types of cells. For example, E-cadherin (or cadherin-1) is mainly expressed on epithelial cells, and N-cadherin (or cadherin-2) is highly expressed on neurons and mesenchymal cells. Liver MSCs also express several cadherins. In particular, at the transcriptome level, MSCs isolated from the liver of patients with fibrosis and cirrhosis express cadherins-2, -4, -6, -11, -22, and -24 (see Table 1). 

Cadherins are the major proteins involved in the formation of adherens junctions. Adherens junctions formed by N-cadherin (or cadherin-2) are important for regulating wound healing [55], cell attachment and migration [56], embryogenesis [57,58], metastasis [59], and also play a key role in cell differentiation and formation of specialized tissues, such as fibrous connective tissues [60,61]. Cadherin-2 is also required for long-term engraftment of hematopoietic stem cells and the restoration of hematopoiesis after bone marrow transplantation [62]. Another important cadherin expressed, among others, in liver MSCs is cadherin-11. It was first identified in mouse osteoblasts, and later its expression was shown on many types of human cells, including mesenchymal stem/stromal cells. The functions of cadherin-11 are diverse. Similar to cadherin-2, this molecule is involved in the formation of adherens junctions and mediates metastasis of tumor cells [63] and synthesis of collagen and elastin, thereby regulating the mechanical properties and contractile function of tissues [64], as well as cell adhesion and proliferation [65]. Both cadherin-11 and cadherin-2 mediate the transformation of fibroblasts into myofibroblasts during the granulation phase of the wound healing process [66]. Cadherins play an important role in the differentiation of MSCs. Accordingly, both cadherin-2 and, most importantly, cadherin-11 are necessary for osteogenesis [67]; cadherin-11 regulates the differentiation of MSCs into smooth muscle cells [68]. Importantly, both cadherins (-2 and -11) are involved in the epithelial-to-mesenchymal transition [69]. During the epithelial-to-mesenchymal transition taking place during fibrosis and carcinogenesis E-cadherin expression is reduced, while cadherin-2 and -11 expression is up-regulated [70].

### 3.5. Cytokeratins and Hepatic Markers

In most studies, it was shown that liver MSCs, in addition to classical mesenchymal markers, also express several hepatic markers, indicating their hepatic commitment. A comparative RT-PCR analysis of primary human hepatocytes, human hepatic stellate cells, HepG2 cell line, and human liver MSCs showed that liver MSCs express mRNA of specific hepatic markers such as albumin and α-fetoprotein [20]. At the protein level, liver MSCs expressed albumin, whereas α-fetoprotein expression was low. At the mRNA level, these cells also expressed other liver-specific markers, such as glucose 6-phopshatase, α-antitrypsin, glutamine synthase, γ-glutamyl transpeptidase, MRP2 transporter, hepatocyte nuclear factor-4, CYP3A4, and CYP1B1, but not CYP2B6, tyrosine aminotransferase (TAT), or tryptophan 2,3-dioxygenase (TDO). The authors believe that the expression of such liver-specific markers is the ultimate proof the hepatic origin of these MSCs [20]. Herrera et al. [21] also showed that liver MSCs were positive for albumin and α-fetoprotein. However, there are differences in the expression of cytokeratins in liver MSCs obtained by different research groups. Najimi et al. [20] showed lack of expression of both cytokeratins 8 and 18, and cytokeratins 7 and 19. At the same time, Herrera et al. [21,71] found that in the absence of expression of cytokeratin 19, liver MSCs express cytokeratins 8 and 18 at a significant level. Other authors observed alternative signs of hepatic commitment of MSCs obtained from human liver: (1) high expression of cytokeratin 19 and hepatocyte growth factor (HGF), low expression of cytokeratin 18, c-Met, and Lgr5, and also lack of albumin expression at the mRNA level [22]; (2) expression of CD26 and cytokeratin 18 by a small subpopulation of cells in the general population of liver MSCs [72].

Our gene expression microarray data show that MSCs isolated from the liver of patients with fibrosis or cirrhosis express several liver-specific genes: ribonuclease RNase A family 4 (*RNASE4*), *CYP1A2*, γ-glutamyl transpeptidase (*GGT1*), and glutamine synthase (*GLUL*)—at a low level; *CYP1B1*—at a relatively high level; a very high level of expression was observed only for the *NNMT* gene encoding the nicotinamide *N*-methyltransferase. Albumin, α-phetoprotein (*AFP*), and hepatocyte nuclear factor-4 (*HNF4A*) were not expressed in our liver MSCs both at the protein level [29] and at the mRNA level. At the same time, expression of the c-Met hepatocyte growth factor receptor was detected by us in 60% of cells in the liver MSC population by flow cytometry [29] as well as at the transcriptome level (see Table 1).

Although the aforementioned markers are considered relatively liver-specific, they can also be expressed in non-liver cells and tissues. For example, proteomic analysis showed the expression of CYP1B1 in bone marrow MSCs [73]. Also, human umbilical cord MSCs constitutively express a whole spectrum of liver markers, such as albumin, α-fetoprotein, cytokeratin-19, connexin-32, dipeptidyl peptidase IV, glucose-6-phosphatase, and claudin [74,75]. The *NNMT* gene is strongly expressed in the human liver [76], and its low level expression is also detected in adipose tissue (subcutaneous and visceral) [77], arteries, muscles, and in various types of mesenchymal cells, in particular, in fibroblasts. Moreover, *NNMT* is expressed at a very low level in the central nervous system and in hematopoietic cells [78]. Several studies have also demonstrated aberrant expression of *NNMT* in several tumors, namely renal cell carcinoma [79], colorectal cancer [80], and gastric cancer [81].

Thus, while MSCs isolated from the human liver by different research groups do differ in individual phenotypic features, they have several common characteristics. In particular, these cells simultaneously express both mesenchymal and hepatic markers, which may indicate their partial hepatic commitment (Table 2). 

At the same time, these cells are different from bipotent hepatic progenitor cells which are human counterparts of rodent oval cells due to the lack of expression of specific markers such as CD34, CD45, and CD117. Also, these liver MSCs are not classic liver mesenchymal cells, i.e., hepatic stellate cells (HSCs). Despite the common phenotypic and morphological characteristics [37], liver MSCs differ from HSCs in the lack of expression of neuronal markers, such as glial fibrillary acidic protein (GFAP), neural cell adhesion molecule (NCAM), neurotrophin-3 (NT-3), CD271 (nerve growth factor receptor or NGFR), etc., which are expressed in HSCs at a high level [82], but are not expressed in liver MSCs [37,83]. Moreover, in several studies, it has been shown that HSCs express CD133 and HLA-DR [84,85], whereas liver MSCs do not express these surface markers. The differences in the expression of markers in liver MSCs obtained by different research groups that were shown in this section can be associated with many factors, including the initial donor material (healthy liver, diseased liver, cadaveric liver, etc.) or details of the method of cell isolation and cultivation.

In general, liver MSCs have much in common with MSCs isolated from other sources. However, careful consideration of data derived by comprehensive screening of cell surface markers expressed by liver MSCs [36] and bone marrow MSCs [86] reveal different levels of the expression of some of them. Liver MSCs exhibit higher expression of such adhesion and cell motility markers as CD49c, CD49e, CD49b, CD63, and CD9, as well as CD46 and CD55 receptors participating in the complement regulation. On the other hand, liver MSCs do not express CD91, CD97, CD130, and epidermal growth factor receptor (EGFR), while bone marrow MSCs do, though at a low level. Considering the level of the uniqueness of liver MSCs, it is necessary to remember that in culture, as with other MSCs, liver MSCs most probably form heterogenous populations, consisting of cells with varying phenotypes. Phenotypic and functional cell variability is typical for the majority of MSC cultures, though its degree may be different [87,88]. For example, single-cell microarray analysis demonstrated high level of variability of the properties of cultured human bone marrow MSCs [89], while umbilical cord [90] or adipose tissue [91] MSCs showed limited variability. This dissimilarity is probably associated with differences in the cell cycle trajectories. The variability of liver MSCs is still difficult to evaluate, because, to our knowledge, no single-cell microarray data are available for them.

## 4. Properties and Therapeutic Potential of Human Liver MSCs

### 4.1. In Vitro and In Vivo Differentiation Potential

As shown in most studies, human liver MSCs can differentiate into cells of the mesodermal lineage. After 3 weeks in culture in osteogenic conditions, i.e., in a growth medium supplemented with β-glycerol phosphate, dexamethasone, and ascorbate 2-phosphate, liver MSCs can differentiate into osteocytes, forming calcium deposits and expressing osteopontin and osteocalcin [21,22,28,34,72]. Only one study reported that liver MSCs are not capable of osteogenic differentiation even after 5 weeks of cultivation, although bone marrow MSCs underwent such differentiation in the same conditions [20].

Unlike osteogenic differentiation, the propensity of liver MSCs for adipogenic differentiation in vitro is somewhat questionable. Some studies show that under standard adipogenic conditions [3] liver MSCs differentiate into adipocytes and accumulate lipid vacuoles [22,28,92], while other works deny this fact [20,21,72]. In our work, we demonstrated that in a population of liver MSCs isolated from the liver of patients with alcoholic cirrhosis, a proportion of cells are able to accumulate lipid droplets 3 weeks after the induction of adipogenic differentiation in a growth medium supplemented with insulin, IBMX (3-isobutyl-1-methylxanthine), dexamethasone, and indomethacin [93] (Figure 2).

Chondrogenic in vitro differentiation of MSCs derived from human liver has been shown in several studies [72].

In addition to differentiation into cells of the mesodermal lineages, human liver MSCs demonstrate the ability to differentiate into cells of other germ layers. In particular, when cultivated in the medium for endothelial differentiation in the presence of vascular endothelial growth factor (VEGF), liver MSCs begin to express endothelial markers such as CD31, CD34, kinase insert domain receptor (KDR or VEGFR-2), CD144 (VE-cadherin), and von Willebrand factor, and at the same time loss of expression of albumin, α-phetoprotein, and cytokeratin 18 and increase of expression of CD105 and CD146 is observed [21]. During the cultivation of liver MSCs in the presence of nicotinamide, the cells lost their elongated fibroblast-like morphology and began to form spheroid clusters morphologically resembling pancreatic islets. The cells in these clusters were positively stained for human insulin and Glut2, which functions as a glucose sensor in pancreatic β-cells [21]. These results demonstrate that liver MSCs are potentially able to differentiate into insulin-producing cells.

Given appropriate cultural conditions, liver MSCs, along with bone marrow and cardiac MSCs, show the ability under specific cultural conditions to differentiate both into neuronal cells expressing GFAP, βIII-tubulin, neuron-specific enolase, and other markers, and into myocytes expressing α-actinin and α-sarcomeric actinin. In both cases, the differentiated progenies of liver MSCs acquire functional activity inherent to nerve cells and myocytes, respectively [34].

Several research groups using different differentiation protocols (Table 3) have demonstrated the ability of human liver MSCs to differentiate into hepatocyte-like cells in vitro [20,21,27,33,34,72,94]. The opposite was reported in just one paper by Porretti et al. [92] claiming that cultured CD105^+^CD73^+^Thy-1^+^CD45^−^CD34^−^ MSCs obtained from the mononuclear fraction of the liver after removal of hepatocytes did not differentiate into the endothelial and hepatic lineages. 

During hepatogenic differentiation, liver MSCs lose their elongated morphology and acquire a polygonal shape. Moreover, hepatocyte-like cells obtained from liver MSCs after hepatogenic differentiation in vitro were shown to down-regulate expression of mesenchymal markers and to up-regulate expression of liver-specific genes and proteins, such as albumin, GATA4, cytokeratins 18 and 19, cytochrome P450, α1-antitrypsin, tryptophan 2,3-dioxygenase, and glutamine synthetase [20,22,27,72]. The hepatocyte-like cells exhibited several traits inherent to mature hepatocytes, albeit at a lower level. For example, liver MSC-derived hepatocyte-like cells secreted albumin and produced urea into the culture medium, and accumulated glycogen [34]. Following in vitro hepatogenic differentiation, liver MSCs began to produce glucose de novo, which was accompanied by up-regulation of the expression of key enzymes, such as pyruvate carboxylase, phosphoenolpyruvate carboxy kinase, and glucose-6-phosphatase. In addition, liver MSC-derived hepatocyte-like cells were able to metabolize ammonium into urea—one of the key functions of mature hepatocytes that involves urea cycle enzymes *N*-acetylglutamate synthase, ornithine transcarbamylase, arginase, and argininosuccinate lyase, whose expression is also up-regulated in the liver MSCs undergoing hepatogenic differentiation. Liver MSCs differentiated into the hepatic lineage up-regulated several main hepatic cytochromes belonging to the three families of cytochromes P450: the CYP1 family (CYP1A1, CYP1A2, and CYP1B1) involved in the metabolism of polycyclic aromatic hydrocarbons; the CYP2 family (CYP2A6, participates in the metabolism of cyclophosphamide; CYP2B6, participates in the metabolism of cyclophosphamide; CYP2C8 and CYP2C9, which plays an important role in ibuprofen metabolism; CYP2E1, involved in metabolism of ethanol and several tobacco-specific nitrosamines); the CYP3 family (CYP3A4 and CYP3A7, the most important drug-metabolizing enzymes). Except for CYP2C8/9, the cytochromes demonstrated increased activity in differentiated liver MSCs [95]. In addition, Baruteau et al. [96] showed that after hepatogenic differentiation of liver MSCs, the resulting hepatocyte-like cells begin to express phenylalanine hydroxylase. The activity of this enzyme in hepatocyte-like cells obtained from liver MSCs was lower than in primary human hepatocytes, but significant enough to make these cells potentially suitable as a treatment for phenylketonuria.

In addition to in vitro hepatogenic differentiation, several authors also presented some evidence of the ability of human liver MSCs to differentiate into hepatocyte-like cells in vivo after transplantation to immunodeficient mice. Najimi et al. [20] transplanted intrasplenic undifferentiated human liver MSCs into 14-day-old uPA^+/+^ severe combined immunodeficiency (SCID) and 6–8-week-old SCID mice (with or without 70% hepatectomy) and showed that after a long enough period (8–10 weeks) human albumin-positive cells but not α-fetoprotein-positive cells were detected in the liver of animals. Moreover, human albumin was also found in the plasma of recipient mice, which, according to the authors, points to the hepatogenic differentiation of human liver MSCs following their homing to the mouse liver. At the same time, the authors showed the presence of albumin expression in undifferentiated liver MSCs at the protein and mRNA levels, and, therefore, these results cannot serve as evidence of the in vivo hepatogenic differentiation of human liver MSCs. However, in a more recent work, the same group of authors [97] found that after intrasplenic injection of human liver MSCs into SCID mice, human cells were engrafted into the mouse liver and showed signs of hepatic differentiation 8 weeks after transplantation. Specifically, human liver MSCs that engrafted the mouse liver lost the mesenchymal marker vimentin and began to express the marker of mature hepatocytes ornithine transcarbamylase [97]. Several other papers demonstrated the ability of human liver MSCs to differentiate into hepatocyte-like cells in vivo after transplantation to immunodeficient mice with CCl_4_-induced hepatic toxicity by analyzing the expression of human albumin in these cells [[22],].

### 4.2. Immunomodulation Properties of Liver MSCs

Based on the immunophenotype of liver MSCs, it can be assumed that much like MSCs obtained from other sources [98,99,100], these cells are likely to manifest immunosuppressive properties. The lack of major histocompatibility complex (MHC) class II expression on the surface of liver MSCs suggests that these cells are weakly immunogenic [72]. On the other hand, similarly to all nucleated cells, they express human leukocyte antigens (HLA) A, B, and C, as well as β2-microglobulin, a component of MHC class I, and therefore, represent targets for cytotoxic T cells [36,72]. According to results from flow cytometry screening of surface markers [36] and our own gene expression microarray data, liver MSCs do not express costimulatory molecules or any other proteins that can induce an immune response. In addition, liver MSCs express regulatory molecules CD46 and CD55 (involved in the inactivation of the C3b and C4b complement proteins), and CD59 (blocks the complement protein C9), which may indicate the presence in these cells of a mechanism that protects them from the complement cascade (Table 1).

Data concerning the expression of CD112 (*NECTIN2*) and CD200 on human liver MSCs are somewhat inconsistent. Dollet et al. [36] did not detect expression of CD112 at the protein level; whereas we showed that its expression is significant in a transcriptome level (our unpublishing gene expression microarray data are present in Table 1). CD112 is predominantly expressed on the plasma membrane of antigen-presenting cells and tumor cells. High-affinity receptors for CD112, such as CD226 and CD112R, are mostly expressed on the surface of T cells. These two receptors, one of them being costimulatory (CD226), and the other one is inhibitory (CD112R), compete for the binding of CD112. Thus, CD112 has a dualistic function relative to T cells; depending on the receptor it binds to, the T cell response may be activated or inhibited. The interaction of CD112 with CD226 leads to costimulation of T cells and an increased T cell response, while the interaction of CD112 with CD112R suppresses T cell reactions [101]. If the expression of CD112 by liver MSCs really takes place, it would additionally highlight their ability to modulate immune responses.

We found that liver MSCs, in contrast to CD112, do not express CD200. On the other hand, Beltrami et al. [22] demonstrated CD200 expression in those cells by cDNA microarray. Also, Dollet et al. [24] detected the expression of this marker on the surface of a limited percentage of liver MSCs (20%–33%) isolated from 3 out of 5 donors. CD200 is an immunoglobulin-like transmembrane surface glycoprotein that is widely expressed by activated T cells, thymocytes, B cells, dendritic cells, and vascular endothelial cells [102]. After binding with its specific receptor (CD200R), CD200 provides inhibitory signals [103], leading to the generation of immunosuppressive and tolerogenic cells that are harmful in the context of tumor progression [104] but helpful in the process of developing tolerance to tissue and organ grafts [105]. Expression of CD200 was detected not only in liver MSCs [36,83] but also in extrahepatic MSCs. CD200 expression on the surface of cultured human bone marrow MSCs is regulated by pro-inflammatory cytokines TNF-α and IL-1β, which play an important role in innate immunity [106]. CD200^+^ human bone marrow MSCs can suppress the secretion of TNF-α by myeloid cells through the binding of CD200 on MSCs with CD200R on myeloid cells, thereby executing an immunomodulatory function [107]. It is possible that the immunomodulatory function of this membrane protein in the case of its expression on liver MSCs is carried out in the same way.

The immunomodulatory/immunosuppressive properties of liver MSCs in the context of their direct influence on immune cells did not yet get enough attention in the literature. It was shown, however, that cocultivation of liver MSCs with mitogen- and alloantigen-stimulated T cells leads to a significant inhibition of proliferation of immune cells [22]. In addition, liver MSCs, to a greater extent than bone marrow and adipose MSCs, suppressed the secretion of IFNγ by immune cells and inhibited the proliferation of CD4^+^ and CD8^+^ T cell populations without affecting the proliferation of CD4^+^CD25^+^Foxp3^+^ Treg cells [108]. The suppression of alloreactive T cells was mediated both by cell-cell contacts between MSCs and lymphocytes and by secreted factors. Blocking of indoleamine 2,3-dioxygenase partially abolished the inhibitory effects of liver MSCs on T cell proliferation. No other mechanisms underlying immunomodulation effects of liver MSCs were yet disclosed. However, it can be speculated that, as with other MSCs, the key role here belongs to cytokines, chemokines, and other secreted factors, especially since the spectrum of biologically active substances produced by liver MSCs is quite extensive [37]. Nuclear factor-kappa B (NF-κB) family of transcription factors participates in the coordination of the inflammatory cascade-related genes expression [109,110]. On the other hand, NF-κB activation mediates the secretion of growth factors and cytokines by MSCs. Recently it has been demonstrated that the telomeric protein Rap1 (repressor/activator protein) is involved in the regulation of the MSC immunomodulation potential. Rap1 is an adapter of the NF-κB modulating enzyme IκB kinase (IKK). Inhibition of Rap1 expression provided substantial reduction of the suppressor activity of MSCs in the mixed lymphocyte culture caused by the suppression of NF-κB and consequent decline of the secretion of cytokines modulating the immune reactions [111]. Since gene expression microarray showed that liver MSCs are characterized by high levels of *NFKB1* and *RAP1A* expression (Table 1), this regulatory mechanism may be involved in the immunomodulation effects of liver MSCs.

Liver MSCs also showed higher ex vivo expression of PD-L1 compared to bone marrow MSCs, which is directly related to inhibition of T cell proliferation and cytotoxic degranulation by liver MSCs [112]. On the other hand, Raicevic et al. [113] detected expression of PD-L1 (CD274) in liver and bone marrow MSCs only after stimulation of the cells with pro-inflammatory cytokines, but not in native cells. Our gene expression microarray showed that MSCs isolated from both fibrotic and cirrhotic liver express CD274 (see Table 1). These controversial results on the expression of PD-L1 are most likely associated with the liver tissues used for isolation of MSCs, namely the existence of an inflammatory process in the original donor material. The inflammatory environment can significantly affect the expression of immunologically functional markers on the surface of MSCs [113,114] and the secretion of cytokines and growth factors by mesenchymal cells, which, in turn, determine the immunomodulatory and regenerative potential of liver MSCs [115].

Based on the existing data, it can be assumed that liver MSCs, similarly to extrahepatic MSCs, possess immunosuppressive/immunomodulatory activity. Further research is needed to verify their effects on the functioning of various individual types of immune cells. Regarding the unique features of liver determining its immunoprivileged status [116], it can also be speculated that liver MSCs may exhibit certain specific traits different from those shown by MSCs isolated from other organs.

### 4.3. Therapeutic Effects of Microvesicles/Exosomes and Conditioned Media Derived from Cultured Liver MSCs

The therapeutic effects of various stem cells, including MSCs, are mostly paracrine. The paracrine action of MSCs is mediated by secreted factors, as well as microvesicles or exosomes. Microvesicles play an important role in intercellular communication, including communication between transplanted stem cells and the recipient’s damaged cells. Microvesicles loaded with various proteins, bioactive lipids, mRNA, and microRNA, act as information carriers inducing phenotypical and functional changes in the recipient cells. The exchange of information between stem cells and damaged cells is reciprocal [117]. Microvesicles derived from liver MSCs can induce proliferation and apoptosis resistance in rat and human hepatocytes in vitro. This effect is ribonucleic acid (RNA)-mediated and requires the internalization of microvesicles into target cells through an α4-integrin-dependent mechanism. It turned out that these microvesicles are loaded with a specific pattern of cellular mRNA, involved in the control of transcription, translation, proliferation, and apoptosis. After intravenous injection of microvesicles obtained from human liver MSCs into rats with 70% hepatectomy, morphological and functional recovery of the liver was observed. This effect was associated with an increase in hepatocyte proliferation in the animal livers, and was abolished when microvesicles were pretreated with RNase, which confirms the role of specific RNA patterns in the hepatoprotective function of the microvesicles [23].

The effects of MSCs on tumor cells are ambiguous. While many studies have shown the supportive role of these cells in tumor growth and progression, others have demonstrated their pronounced antitumor effects. As in the case of regenerative properties of MSCs, their antitumor effects can also be mediated by paracrine signals. Microvesicles derived from liver MSCs exhibit antitumor effects in vitro and in vivo [118]. Specifically, these microvesicles inhibit proliferation and induce apoptosis of the HepG2 cell line and primary cultures of hepatocellular carcinoma (HCC) obtained from tumor samples of patients with diagnosed HCC after resection. The effects are dose-dependent and require internalization of microvesicles into tumor cells, which is α4 integrin- and CD29-dependent. Intratumoral injection of liver MSC-derived microvesicles in a xenogeneic mouse tumor model resulted in a significant decrease in the size and weight of the tumor compared to the controls. At the same time, the proliferation of HepG2 cells inside the tumor decreased, and the level of their apoptosis increased. These microvesicles exert similar antitumor effects in vitro and in vivo on other tumors, including SupT1 lymphoblastoma and DBTRG-05MG glioblastoma [118]. Many studies have shown that the transfer of specific miRNAs that suppress tumor growth, inhibit tumor angiogenesis, migration, and/or invasion into tumor cells plays an important role in the antitumor effect of microvesicles derived from various MSCs [119]. The molecular composition and specific features of the microvesicles/exosomes strictly correlate with the origin of the cells from which they are derived. Therefore, microvesicles/exosomes obtained from MSCs of different origin can have different effects on the same recipient cells [120]. Collino et al. [121] showed that microvesicles derived from liver MSCs, as well as those from bone marrow MSCs, contain several functionally active miRNAs, such as miR-24, miR-222, miR-99a, miR-125b, miR-100, miR-31, miR-19b, miR-16, miR-594, and miR-125a. In addition, liver MSC-derived microvesicles contained several specific miRNAs that were absent in bone marrow MSC-derived microvesicles, namely miR-650, miR-95, miR-7, and miR-204. Fonsato et al. [118] found that among many miRNAs present in the liver MSC-derived microvesicles, miR451, miR223, and miR31 display the most pronounced antitumor effects on the hepatoblastoma cell line HepG2. Several other miRNAs that are present in microvesicles may also have antitumor effects. Significant antitumor effects in vitro and in a xenograft mouse hepatoblastoma model were observed for microvesicles obtained from adipose-derived MSCs that were modified with miR-122; however, analogous effects were not detected for corresponding naïve microvesicles. Moreover, such miRNA-modified microvesicles increased the sensitivity of HepG2 cells to chemotherapy drugs [122]. Similar anti-hepatoblastoma effects were also shown for bone marrow MSC-derived microvesicles; such non-modified microvesicles inhibited the progression of the cell cycle, causing arrest in G0/G1, and induced apoptosis in HepG2 cells. Intratumoral administration of these microvesicles into SCID mice with an established tumor that was generated by subcutaneous injection of HepG2 cells significantly inhibited tumor growth [123].

The conditioned medium obtained from liver MSCs contrasted with the conditioned medium from bone marrow MSCs in that it inhibited the growth of HepG2 in vitro and in vivo, and induced apoptosis of tumor cells. According to Cavallari et al. [124], the Lefty A protein, which is secreted by liver MSCs but is not expressed in bone marrow MSCs, plays an important role in this tumor inhibition effect. The conditioned medium from liver MSCs also induced a significant decrease in Nodal expression in hepatoma cells at the protein and mRNA levels, which indicates the important role of Nodal signaling in the progression of the HepG2 hepatoma cell line [124]. Lefty is a natural inhibitor of Nodal, whose function is to tightly control Nodal signaling during fetal development. However, Lefty is absent in cancer cells, which leads to uncontrolled tumor growth [125]. Therefore, inhibition of the Nodal pathway (using conditioned media and/or microvesicles containing Lefty, among other approaches) in tumors that primarily require Nodal activation to progress may represent an important therapeutic strategy.

As has been shown for extrahepatic MSCs, liver MSCs can also have a stimulating effect on tumor cells. Kellner et al. [30] showed that along with bone marrow and adipose tissue-derived MSCs, liver MSCs co-cultured with the chronic myelogenous cell line K562 cause extensive clonal proliferation of tumor cells.

Thus, the above-mentioned data demonstrates that both microvesicles/exosomes and conditioned media obtained from various MSCs, including liver MSCs, can serve as a useful, effective, and hopefully safe instrument in the therapy of cancer diseases. At the same time, more research is required in this area, since the exact mechanisms of antitumor action of microvesicles/exosomes, conditioned media, and their individual components are unknown. Also, the reasons, why microvesicles/exosomes and/or conditioned media from some MSCs have a stimulating effect on particular types of tumors, while microvesicles and/or conditioned media obtained from other types of MSCs exercise inhibitory effects, need to be established. In this context, both the origin of MSCs and the origin of the tumor probably play an important role, and the choice of the optimal source of microvesicles as a therapeutic agent is an important scientific task [126].

### 4.4. Biodistribution of Liver MSCs after Transplantation

Human liver MSCs show a high regenerative potential following transplantation into immunodeficient mice in various models of liver pathologies. At the same time, regardless of the route of administration (intravenous, intrasplenic, or intraperitoneal), human liver MSCs demonstrate the ability for homing to and engraftment in the damaged mouse liver. In most cases, the cells persist in the liver of recipient mice for quite a long time (30–56 days) and even differentiate into hepatocyte-like cells (as described in the previous section).

The biodistribution of human liver MSCs transplanted into immunodeficient mice using various routes of administration has been studied, albeit incompletely. Seven days after intravenous administration of human liver MSCs into SCID mice with induced liver damage, human cells were detected in the liver parenchyma [21]. It is likely that just a small number of transplanted cells reached liver, since upon intravenous administration they also migrate and are distributed to other organs. Really, one day after the injection of human liver MSCs into the tail vein of NOD/SCID/IL-2Rg(null) mice with CCl_4_-induced fibrosis most of the cells were observed in the lungs. Over time, the amount of human DNA in the lungs of mice decreased, while it increased in other organs, in particular, in the liver. Following the accumulation of human cells in the murine liver by day 7 after transplantation, the amount of human DNA did not decrease up to day 56, indicating that human cells effectively engrafted in mouse liver. However, despite effective engraftment, transplanted human cells did not affect the recovery of mice from fibrosis. The lack of the positive effect was probably due to the insufficient number of initially injected cells (500,000/mouse/injection) [].

Human liver MSCs were detected in the liver 10 min after intrasplenic administration into SCID mice without liver damage. Human cells were mainly detected near the portal tract. 7 days after transplantation, some human cells engrafted in the mouse liver parenchyma, although the percentage of engrafted cells was less than 1%. Proliferation of human cells in the liver was observed in mice 2–4 weeks after the injection of human liver MSCs followed by 20% partial hepatectomy. During this time human MSCs differentiated into hepatocyte-like cells, highlighting the involvement of these cells in the reaction of the liver towards hepatectomy [97].

### 4.5. Therapeutic Effects of Liver MSCs

Liver fibrosis is an unsolved medical problem. Fibrosis develops in response to chronic liver damage (e.g., viral damage caused by hepatitis B or C) and, in the absence of proper control, proceeds to cirrhosis, and the end-stage of the liver disease [127]. The treatment of fibrosis aims to achieve four main goals: elimination of the cause, suppression of inflammation in liver tissue, suppression of the activation of HSCs, and enhancement of the destruction of the extracellular matrix. The most effective way to prevent the formation of fibrosis is to eliminate its cause, but this is not always possible [128]. Hepatic stellate cells (Ito cells) (HSCs) are the main producers of the extracellular matrix in the liver. HSCs are activated during chronic liver damage and turn into myofibroblasts with fibrogenic properties [129]. Over the past few years, many studies have reliably shown that pharmacological deactivation and/or removal of myofibroblasts, as well as the use of antagonists of profibrogenic factors, leads to a significant reduction of fibrosis [130]. Stem cell therapy may represent a promising approach in the treatment of liver fibrosis [131]. In particular, anti-fibrotic activity in vitro and in vivo was shown for many types of MSCs. Human umbilical cord and bone marrow MSCs effectively suppress the proliferation of HSCs, inducing their apoptosis, and significantly reduce the production of the main pro-fibrogenic factors such as TGF-β1 and enhance the production of anti-fibrogenic cytokines such as HGF and IL-10 by Ito cells [132,133]. Human liver MSCs also showed anti-fibrogenic potential in vitro and in vivo. Liver MSCs significantly reduce proliferation and inhibit the adhesion of HSCs when co-cultivated in the Transwell assay. The same effect was demonstrated for the conditioned medium obtained from liver MSCs. Moreover, both autologous and allogeneic liver MSCs suppressed the proliferation and decreased the adhesion of HSCs. Suppression of the proliferation of HSCs was not caused by cell death, but rather by the cell cycle arrest at the G0/G1 phase. When HSCs are activated, secretion of type I collagen increases. Following cocultivation of liver MSCs with activated HSCs, the level of synthesis of procollagen type I significantly decreased, and the secretion of anti-fibrotic factors, such as HGF, IL-6, MMP1, and MMP2, increased. The most powerful anti-fibrogenic effect was observed for HGF since blocking it with antibodies terminated the inhibitory effects of liver MSCs and the conditioned medium obtained from liver MSCs [134]. During intrahepatic transplantation of human liver MSCs into young rats which were chronically treated with phenobarbital and CCl_4_, the expression of genes associated with fibrosis, including α-SMA, Col1α1, Loxl1, and TIMP-1, decreased in 3 out of 6 rats. Protein markers of myofibroblasts, including α-SMA and fibronectin, which are up-regulated after CCl_4_-induced liver fibrosis in rats, were also significantly reduced after transplantation of human liver MSCs [134].

In the mouse model of fulminant liver failure, which implies 100% animal mortality during the first 8 h after induction, the transplantation of human liver MSCs by various routes led to a significant increase in animal survival [135]. Specifically, a single intraperitoneal injection of human cells 30 min after the injury resulted in 77% survival, whereas a combination of a 6-fold intravenous and a single intrahepatic injection led to 100% survival. Also, intraperitoneal injection of conditioned medium obtained from human liver MSCs increased survival by up to 80%. Upon all three above-mentioned routes of administration, human MSCs engrafted the mouse liver and persisted there for up to 21 days, providing a stimulating effect on the regeneration of the damaged liver. Necrosis and apoptosis were significantly reduced following 3 days, and normal liver morphology was restored in 21 days. The conditioned medium obtained from human liver MSCs had a similar effect on the regeneration of the liver during fulminant liver failure. Such conditioned medium contains several growth factors/cytokines potentially involved in the protection and regeneration of the liver, namely IL-6, IL-8, VEGF, HGF, and macrophage-stimulating protein (MSP). Human HGF exerted the most pronounced hepatoprotective effect, which was shown using blocking antibodies that led to suppression of the protective effect of the conditioned medium [135].

When the conditioned medium obtained from human liver MSCs was intraperitoneally administered to mice immediately after 70% hepatectomy, it significantly increased the proliferation of hepatocytes, albeit had no effect on the metabolic function. Simultaneously, the up-regulation of genes such as TNF-α, HGF, PCNA, and TGF-β, as well as the pro-angiogenic factor genes including VEGF-A, VEGF-R1, VEGF-R2, and Ang-1 was observed in the mouse liver [136]. These data demonstrate that not only the cells themselves but also the conditioned medium display hepatoprotective functions. 

In addition to stimulation of liver regeneration after injury, human liver MSCs also can restore the metabolic function in various metabolic disorders. In particular, human liver MSCs are capable of specific conjugation of bilirubin after intraportal injection into Gunn rats. Gunn rats are a model of human Crigler–Najjar syndrome type I, which is characterized by the lack of bilirubin conjugation function. The liver metabolic function in rats reached normal values within 3 months after transplantation of human cells and persisted up to 6 months [137].

Though the mechanisms underlying the therapeutic effects of liver MSCs remain poorly understood, available data suggest that their ability to regenerate the injured liver can be accomplished in three ways (Figure 3): (1) through anti-inflammatory activity and immunomodulation; (2) through anti-fibrotic action; (3) through MSCs’ ability to differentiate into the hepatocyte-like cells.

High efficacy of human liver MSCs in animal disease models indicates the involvement of species-independent mechanisms. Most chronic liver diseases, as well as hepatocarcinogenesis are associated with enhanced oxidative stress [138,139]. It has been repeatedly shown that transplantation of different MSCs [140] or MSC-derived exosomes [141] into animals with injured liver induced significant anti-oxidative protection. The antioxidant action of MSCs is primarily associated with high constitutive expression and secretion of enzymes regulating the oxidative stress, including superoxide dismutase 1 (SOD1) [142], SOD3 [143], glutathione S-transferase P (GSTP1) [144], glutathione peroxidases [145] (Table 1). Delivery of glutathione peroxidase 3 by MSCs derived from human induced pluripotent stem cells and transplanted into the damaged liver suppressed the oxidative stress and, consequently, inhibited hepatocyte aging and restored liver tissue [146]. These results open the possibility to create engineered MSCs with enhanced therapeutic characteristics, including gene-engineered MSCs [147]. Importantly, human liver MSCs express high levels of anti-oxidative enzymes (Table 1) and this probably contributes to their therapeutic efficacy. 

Oxidative stress observed during chronic liver diseases is associated with hepatocyte mitochondrial dysfunction. In addition to their role in the genesis of oxidative stress, mitochondria regulate hepatic lipid metabolism. It has been demonstrated that patients with alcohol-associated and non-associated liver diseases have mitochondria with damaged ultrastructure and substantially lower level of the adenosine triphosphate synthesis [148]. In the animal model of the non-alcoholic fatty liver disease (NAFLD) intravenous infusion of mitochondria isolated from healthy cells caused substantial reduction of the oxidative stress and depletion of liver lipids [149]. Several studies presented data suggesting that mitochondria transfer from MSCs to the damaged cells may be one of the key mechanisms of the regeneration-promoting effects of MSCs [150]. It has been proven that horizontal transfer of mitochondria from MSCs occurs not just in vitro, but in vivo [151]. It is a normal physiological process serving to substitute damaged organelles for functionally active ones [152]. Moreover, traffic of mitochondria goes in two directions and their transfer from damaged cells to MSCs signals danger and promotes MSC therapeutic activity [153]. Transfer of mitochondria from transplanted MSCs to damaged liver cells has not been demonstrated. However, taking into account the crucial role of mitochondria in hepatocyte physiology and the curative power of transplanted MSCs in liver diseases, this mechanism is likely to be involved in the therapeutic action of transplanted MSCs. 

Since liver MSCs, similarly to mesenchymal stem cells from other tissue sources, have a high proliferative potential in culture, are able to differentiate into a plethora of cell types, and also effectively engraft the organs of recipient animals after transplantation and persist for a prolonged period of time, the safety of these cells in the context of malignant transformation represents an important topical question. Two studies showed that liver MSCs, along with bone marrow and cardiac MSCs did not form colonies in soft agar and neoplasia after subcutaneous injection into immunodeficient mice [34,154]. Furthermore, during long-term cultivation, liver MSCs acquired several chromosomal aberrations and cytogenetic abnormalities, but even cells with an unstable genome did not show signs of malignant transformation in vitro and in vivo. All liver MSCs achieved cell senescence and cell cycle arrest in culture, which gives evidence of their safety [154].

### 4.6. Clinical Trials of Human Liver MSCs

The above-mentioned studies of the therapeutic potential and safety of human liver MSCs served as a proof-of-concept basis for trials on the transplantation of these cells to individual patients with various metabolic disorders [155]. The first implantation of allogeneic human liver MSCs was performed on a 3-year-old patient who suffered from severe ornithine carbamoyltransferase deficiency. The cells were injected intraportally (0.9 × 10^9^ cells) and constituted 0.75% of the theoretical liver hepatocyte mass. Analysis of the biopsy material following 3 and 6 months after injection showed that the number of transplanted cells in the recipient liver was 3–5% of the theoretical liver hepatocyte mass. These theoretical calculations give evidence of the proliferation of the injected cells and of the repopulation of the recipient’s liver with these cells. The patient also showed improvement in several clinical indicators (decompensation episodes with milder level of NH_3_) after transplantation of allogeneic liver MSCs [156]. An analysis of the biodistribution of allogeneic liver MSCs labeled with 111-Indium DTPA (diethylenetriaminopentaacetic) radiotracer after infusion through the portal vein in a 17-year-old patient with glycogenosis type 1A showed that the injected cells were homogeneously distributed throughout the liver and were not detected in any other organs [157]. Also, the analysis of the biodistribution of allogeneic liver MSCs labeled with 111-Indium DTPA after intravenous administration in a 31-year-old male patient who suffered from severe hemophilia showed that the cells initially accumulated in the lungs and then homed into the liver and to a joint afflicted with hemarthrosis. After cell therapy, a temporary decrease in the patient’s need for factor VIII and a much smaller number of bleeding episodes during physical exercise were observed, despite the lack of rise in circulating factor VIII [158].

There are currently three ongoing clinical trials that use liver MSCs or adult-derived human liver stem/progenitor cells (HepaStem) for the treatment of patients with liver diseases, one in adult patients with acute-on-chronic liver failure (NCT02946554), second in adult patients with non-alcoholic steatohepatitis (NCT03963921), and the third in pediatric patients with metabolic diseases, such as urea cycle disorders (NCT03884959) (Table 4). The pediatric Phase II trial is in the patient recruitment stage; it follows the Phase I/II trial on liver-derived mesenchymal stem cells in pediatric liver-based metabolic disorders, the results of which were recently published [159]. The purpose of the initial trial was to assess the safety of intraportal infusion of allogeneic liver MSCs (HepaStem) 6 months after transplantation. The secondary aim was to assess the safety and preliminary efficacy 12 months after transplantation. 15 pediatric patients with urea cycle disorders and 6 pediatric patients with Crigler–Najjar syndrome took part in the trial. In both groups, liver MSC infusion showed overall safety and good tolerability [159].

Transplantation of MSCs isolated from different sources (cell therapy with MSCs) is a popular trend in regenerative medicine. Clinical trial data usually demonstrate safety, tolerability, and beneficial clinical outcomes. However, several studies demonstrated that MSCs-based therapy may induce thrombosis [160,161], probably due to the pro-coagulant activity of MSCs expressing the tissue factor (TF) on their surface [162,163]. Human hepatocytes [164] and human liver MSCs [165] also possess the pro-coagulant activity. MSCs-induced thrombosis is dose-dependent and more pronounced after infusion of high doses of MSCs [166]. To avoid thrombosis, most authors suggest anti-coagulant medications, such as heparin. Though anti-coagulants really attenuate thrombosis-related complications [165], they can negatively affect human liver MSCs’ therapeutic efficacy. It was shown that homing of the progenitor cells and stimulation of liver regeneration are enhanced by TF-associated pro-coagulant activity [167,168]. Anti-coagulants, on the contrary, interfere with the homing of transplanted cells within tissues by targeting the CXCR4/SDF-1 axis [169]. As shown in a clinical trial conveyed by Coppin et al. [170], infusion of liver MSCs into patients with decompensated acute liver cirrhosis in the absence of anti-coagulant therapy may be safe in case of close control over the platelets counts and the level of fibrinogen. The results described above are getting more support from clinical evidence. Thus, recently HepaStem presented the preliminary report of a clinical trial involving infusion of liver MSCs in patients with acute-on-chronic liver failure or acute decompensation. It was reported that infusion of a high dose of liver MSCs (3.5 × 10^6^ cells/kg) caused enhancement of blood coagulation in 2 of 3 patients, while repeated injections of lower doses of HepaStem (0.5 × 10^6^ cells/kg/infusion) (with an interval of 1 week) did not induce pro-coagulant effects [171].

Based on the encouraging results concerning the main characteristics and therapeutic potential of human liver MSCs, it is likely that these cells can provide effective tools to treat liver diseases. Due to inherent hepatic commitment, liver MSCs can be more suitable for this purpose than their more easily available counterparts, such as the adipose-derived MSCs. Access to autologous liver MSCs is problematic, but since recent clinical trials demonstrated safety and efficacy of allogeneic liver MSCs, it is possible to use post-mortal material or imperfect liver grafts as the source of cells for transplantation.

## 5. The Origin of Mesenchymal Stem Cells in the Liver

The cellular composition of the liver was described in detail long ago, and, until recently, the presence of a specific population of liver progenitor cells had been questionable. This is due to the fact that in most of the early works aimed at studying the regeneration of the liver, it was reliably proved that the preexisting fully differentiated hepatocytes and cholangiocytes have the ability to quickly restore the liver tissue after loss of its part or after injury [172]. Liver regeneration after partial hepatectomy is indeed mainly mediated by hepatocytes and represents a fast and highly regulated process that results in the restoration of initial organ mass, its architecture and functions [173]. However, as it became known later, in the case of chronic liver damage (for example, chronic viral hepatitis, sub-massive necrosis, and non-alcoholic fatty liver disease) or blockade of the regenerative potential of hepatocytes, specific resident progenitor/stem cells are involved in the organ repair process. These cells were found in rodent livers in the middle of the last century [174,175], and are called “oval cells” due to the oval shape of their nuclei. The presence of similar cells in the human liver was shown in the 1990s, and they are conventionally called hepatic progenitor cells (HPCs) or liver progenitor cells (LPCs) [176]. Like the rodent oval cells, human liver progenitor cells have an oval-shaped nucleus and are localized in the periportal region. The incidence of these cells in the liver is very low and constitutes approx. 1 LPC per 10,000 liver cells. Hepatic progenitor cells are bipotent, they can differentiate into hepatocytes and cholangiocytes and simultaneously express markers of bile duct cells (cytokeratin 19, A6, and OV6) and fetal hepatocytes (α-fetoprotein and albumin) [177].

In addition to the liver progenitor cells, stem and progenitor cells of the hematopoietic, endothelial, epithelial, and mesenchymal lineages were also identified in healthy and diseased human livers. Within the mononuclear liver cell compartment, the epithelial progenitors [EpCAM^+^CD49f^+^CD29^+^CD45^−^] constituted 2.7–3.5%, whereas hematopoietic (CD34^+^CD45^+^), endothelial [KDR^+^CD146^+^CD45^−^], and mesenchymal [CD73^+^CD105^+^CD90^+^CD45^−^] stem cells and progenitors collectively accounted for a very small fraction (0.02%–0.6%). The frequency of MSCs in diseased and healthy livers was similar and comprised 0.6% [92]. Progenitor cells phenotypically corresponding to mesenchymal stem cells were first isolated from an adult human liver in 2006 by Herrera et al. [21]. The authors found that the frequency of these human liver MSCs in an adult human liver is 5.33±1.75 MSCs per 0.5×10^6^ hepatocytes, which corresponds to one MSC per 94,000 hepatocytes. In 2007, Najimi et al. [20] isolated and characterized adult-derived human liver stem/progenitor cells that simultaneously expressed both mesenchymal and hepatic markers and were able to effectively differentiate into hepatocyte-like cells in vitro and in vivo. Following that, several more groups isolated MSCs from both healthy and diseased adult human livers. However, the question of the origin of mesenchymal stem/progenitor cells in the liver remains open till now.

There is a hypothesis that pericytes resting in the walls of capillaries, venules, and arterioles can be a source of mesenchymal stem cells [178]. Main confirmation for this hypothesis came from the studies that analyzed the expression of a set of surface markers and assessed the differentiating potential of the isolated cells. However, mesenchymal stem cells can be isolated not only from tissues containing a network of small blood vessels, but also from the walls of large blood vessels that do not contain pericytes, in particular, the aorta, the saphena vein, and the umbilical cord vein [179]. Thus, adventitial cells of human arteries and veins may represent a different source of MSCs [180]. Since it is still highly likely that MSCs originate from pericytes, it can be assumed that, given the high vascularization of the liver, MSCs that reside in this organ also originate from pericytes. It is generally accepted that hepatic stellate cells, or Ito cells, are pericytes of the liver [181]. Even though HSCs express neural crest markers in addition to mesenchymal markers, their developmental origin is a sheet of mesoderm [182,183], which is similar to the origin of mesenchymal stem cells in embryogenesis [184,185]. Moreover, human HSCs and human liver MSCs have a very similar phenotype; these two cell types demonstrate a similar expression profile of mesenchymal, hematopoietic, and extracellular matrix markers [37]. Also, progenitor properties of HSCs, for example, their ability to differentiate into hepatocytes, were shown in many studies [186,187]. Many authors consider HSCs to be liver-resident MSCs [188]. Nonetheless, liver MSCs obtained by other authors significantly differ from HSCs, mainly due to the lack of expression of hepatic stellate cell markers such as GFAP, NCAM, NT-3 [37,82].

Some researchers argue that hepatic pericytes are a separate subpopulation of mesenchymal cells different from hepatic stellate cells. Gerlach et al. [30] isolated pericytes from fetal and adult human livers. Isolated cells expressed both pericyte and mesenchymal markers, such as CD146, NG2, desmin, CD90, CD140b, and vimentin, but did not express the hepatic stellate cell marker GFAP. Immunohistochemical analysis of fetal and adult human liver tissues showed that CD146-positive pericytes were negative for α-SMA and were localized in the periportal region, but not in the pericentral region or within the space of Disse, where HSCs are located. These cells easily differentiated to osteogenic and myogenic lineages and weakly differentiated to adipogenic and chondrogenic lineages in vitro. Similar cells, namely pericytes with the CD146^+^CD34^−^CD56^−^CD45^−^ phenotype, were also found in mouse liver. Mouse CD146^+^ hepatic pericytes expressed markers of adult mesenchymal stem cells (CD73, CD105, and CD44), as well as other markers (PDGFRβ, NG2, and vimentin), and easily differentiated into adipogenic, osteogenic, and chondrogenic lineages. As in the case with human hepatic pericytes, they did not express HSC markers αSMA and GFAP [189]. It is possible that hepatic pericytes other than HSCs may in fact be the source of liver MSCs, i.e., these cells are resident liver mesenchymal stem cells. This hypothesis is supported by the evidence that liver MSCs express the pericyte surface marker CD146, though to a variable degree [21,71].

Circulating bone marrow progenitor cells may represent a different source of liver MSCs. In response to damage and/or inflammation, bone marrow progenitor cells, including MSCs [190], are released into the bloodstream and travel towards the sites of injury. Pedone et al. [191] showed the importance of bone marrow hematopoietic stem cell recruitment in liver regeneration in mice after 20% and 70% hepatectomy. The authors found that blocking the process of mobilization of bone marrow hematopoietic stem cells into the liver markedly impaired the regeneration of the organ after surgery. Actually, the recruitment of bone marrow cells to the liver, their fusion with hepatocytes, and the subsequent proliferation of such hybrid cells that occurred more rapidly and intensively than the proliferation of hepatocytes, were crucial for the recovery of the liver. Recruitment of bone marrow hematopoietic stem cells to the liver has also been shown in humans after adult-to-adult living donor liver transplantation [192]. After transplantation of a part of the liver obtained from female donors into male patients, vessels formed in the engrafted tissue contained not only donor cells but also recipient cells. Moreover, in female liver grafts, circulating CD34-expressing cells were observed among the Y-probe-labeled, recipient-derived cells, which suggests that circulating bone marrow progenitor cells are involved in the engraftment of donor tissue and regeneration of the liver [192]. Mobilization of endogenous bone marrow MSCs into fibrous liver and their participation in the regenerative process were demonstrated in a mouse model of CCl_4_-induced fibrosis. Mobilization of MSCs from bone marrow to liver was mainly regulated by the SDF-1α/CXCR4 chemotactic axis. The migration of MSCs from bone marrow did not begin at the moment of liver injury, but only after the balance of SDF-1α expression between liver and bone marrow significantly shifted towards an increase in the expression of the factor in liver [193]. Nishiyama et al. [194] showed that the number of CD11b^+^ Kupffer cells/macrophages recruited from the circulation and the bone marrow significantly increased in the liver of mice after partial hepatectomy and greatly accelerated regeneration. Liver sinusoidal endothelial cells (LSECs) also play an important role in the regeneration of the liver [195]. DeLeve et al. [196] have recently proved that not only mature resident LSECs but also progenitors of sinusoidal endothelial cells recruited from the bone marrow are involved in liver regeneration, and that the role of the progenitor cells in this process is more pronounced. Thus, the mobilization of various types of stem/progenitor cells from both the bone marrow and the circulation during the pathological process in the liver has been consistently proven.

High quantities of MSCs are present in primary hepatic tumors [197], as well as in hepatic metastases of colorectal cancer [198]. Yan et al. [199] were the first to demonstrate the presence of MSCs in hepatocellular carcinoma (HCC) and adjacent tumor-free tissues. MSCs isolated from HCC and normal tumor-free liver tissue differentiated to adipogenic and osteogenic lineages in vitro, expressed mesenchymal markers CD29, CD73, CD166, CD90, and CD105, and did not express CD45, CD14, CD144, CD31, and HLA-DR. Immunocytochemical analysis showed that tumor liver MSCs and non-tumor liver MSCs are also positive for vimentin, α-SMA, and N-cadherin, and negative for epithelial markers cytokeratin 18 and E-cadherin [199]. An immunohistochemical analysis of HCC samples and the adjacent tumor-free liver tissue showed that in normal tissue STRO-1-positive cells (MSCs) were localized in liver sinusoids or veins, and their frequency was very low, except for areas with high inflammation. The tumor stroma contained a much larger number of STRO-1-positive cells [197]. HCC-associated MSCs contributed to the progression of the tumor, enhanced the formation of the tumor spheres, increased the expression of cancer stem cell markers such as CD90 and CD13 in hepatocarcinoma cells in vitro, and contributed to the liver cancer stemness, including tumorigenicity and metastases in vivo [197,199,200]. It is still unknown whether the HCC-associated MSCs are BM-derived hepatotropic cells, which subsequently acquire specific gene expression, or whether they are resident liver cells homed in the liver tissue during organ development. The factors secreted by hepatocellular carcinoma cells can induce increased migration of MSCs in vitro and, with high efficiency, stimulate the recruitment of these cells to the tumor stroma in vivo [201,202]. Since the highest enrichment of MSCs is observed in tumors with extensive inflammation [197], it can be assumed that recruitment of MSCs from the bone marrow and circulation in response to infection/inflammation is taking place. This assumption is also supported by the fact that the peripheral blood of HCC patients contains elevated levels of bone marrow endothelial progenitor cells capable of migrating to the tumor and contributing to its progression [203]. Based on these data, Hernanda et al. [204] suggested that MSCs in hepatic cancer have a dualistic origin with simultaneous recruitment of both the resident MSCs that constantly persist in the liver and the MSCs from circulation.

Considering the above-mentioned results, it seems possible that at least in the damaged liver the pool of mesenchymal stem cells largely consists of migrated bone marrow MSCs, recruited as a reaction to injury/inflammation. Though liver MSCs, unlike bone marrow MSCs, express a wide range of liver-specific genes, their hepatic commitment can be explained by the influence of the local environment, which reflects the importance of specific niche conditions in determining the fate of stem/progenitor cells [205,206,207]. Nevertheless, the possibility of existence of resident MSCs that are constitutively present throughout the liver development as a specific subpopulation and are different from such recognized resident mesenchymal liver cells as HSCs cannot be excluded. 

There is one more hypothesis exists regarding the origin of MSCs in the liver. Not so long ago, it was shown that under certain conditions mature hepatocytes dedifferentiate into oval cells, which can in turn proliferate and redifferentiate into hepatocytes in vivo [208]. Luo et al. [209] obtained adult rat hepatocyte-derived mesenchymal-like stem cells (arHMSCs) in culture from rat mature hepatocytes without the use of toxic chemicals or reprogramming and redifferentiated them into functionally active hepatocyte-like cells suitable for drug testing. Hepatocyte-derived mesenchymal-like stem cells obtained in this way expressed mesenchymal markers CD29, CD44, CD90, α-SMA, actin, and vimentin, and did not express liver-specific markers asialoglycoprotein receptor (ASGPR) and albumin, as well as the hematopoietic marker CD45 [209]. Considering the fact that in most studies human liver MSCs were obtained from the parenchymal fraction of the liver and even from frozen hepatocytes, and also taking into account the unique features of hepatocytes regarding their potential for dedifferentiation in vitro and in vivo, it seems plausible that liver MSCs may be the descendants of mature hepatocytes that underwent epithelial-to-mesenchymal transition in the process of dedifferentiation.

## 6. Conclusions

Data reviewed above show that application of the procedures common for MSCs isolation from different tissues to human liver tissue obtained from liver biopsies, autopsies, livers withdrawn from patients before transplantation or pre-transplantation liver grafts yields cells sharing some crucial characteristics with MSCs from other sources, but also exhibiting several unique features. MSCs can be isolated from liver graft perfusates as well. All cultures of liver MSCs established in this way are heterogenous and in this respect are similar to other MSC cultures. It is still not quite clear how different among themselves cultures derived from the above listed sources of human liver MSCs are.

In the culture conditions human liver MSCs acquire typical MSC morphology. Their phenotypes and gene expression profiles, though varying among published reports, mostly match those of classical MSCs from bone marrow or fat, but at the same time partly overlap with those of hepatocytes. Enhanced expression of liver-specific genes can indicate that liver MSCs are inherently committed to hepatic lineage. Regarding the phenotypes and gene expression profiles, liver MSCs are clearly different from known bipotent hepatic progenitor cells as well as from hepatic stellate cells. 

It has been repeatedly shown that human liver MSCs can be induced to differentiate into cells of mesodermal lineage. Remarkably, compared to osteogenic, chondrogenic, and myogenic differentiation, their ability to adipogenic differentiation is somewhat hindered. In addition, human liver MSCs can differentiate into cells derived from the ectoderm (neuronal cells) or endoderm (pancreatic beta-cell-like cells, hepatocytes). Hepatogenic differentiation of human liver MSCs may prove to be very important from the practical point of view, because potentially it can solve the problem of the lack of high-quality human hepatocytes for cell therapy of liver diseases, liver-tissue engineering, drug and toxic substances testing, and basic research. Unlike human hepatocytes, human liver MSCs can be easily produced in large quantities in vitro and then induced to differentiate into hepatocytes. Notably, other types of human MSCs, including MSCs isolated from bone marrow, adipose tissue, and umbilical cord Wharton’s jelly also can undergo hepatogenic differentiation. However, liver MSCs, despite difficulties of access to human liver tissue, perhaps are the best choice, because they are originally committed to hepatic lineage. The accurate comparison of hepatocytes derived from human liver MSCs and other MSCs has not been done yet. 

Human liver MSCs and MSCs isolated from other tissues share several features valuable from the clinical perspective, including paracrine modulation of immune response and inflammation. However, again thorough valuation of the immunomodulatory, anti-inflammatory, and pro-inflammatory activity of human liver MSCs still needs to be carried out. The role of these cells in the functioning of the healthy liver and/or in the development of the pathological process is obscure. It has so far been studied to some extent only in the context of the development of liver tumors. More research into the properties of liver MSCs is required to ascertain whether a resident subpopulation of cells exists in the liver that possesses specific properties and perform specific functions characteristic only for these cells. Currently it is a matter of debate what liver or extrahepatic cells are predecessors of liver MSCs maintained in vitro. Studies in this area are likely to reveal new insights into some crucial areas of cell biology, such as interactions between parenchymal and stromal cells, epithelial-to-mesenchymal, and mesenchymal-to-epithelial transition and liver-tissue regeneration. Anyway, at present studies of liver MSCs represent fascinating areas of fundamental research with a potential for important practical applications.

## Figures and Tables

**Figure 1 cells-08-01127-f001:**
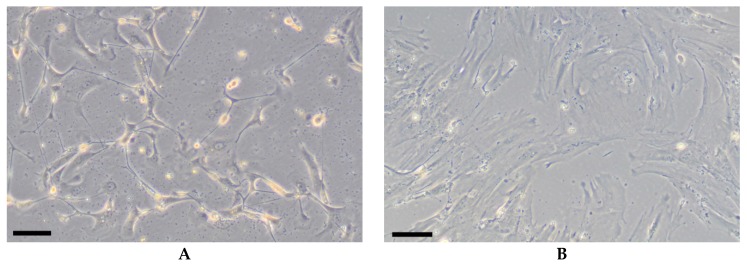
Morphology of liver MSCs isolated from liver of patients with fibrosis. **A**—liver MSCs 5 days after isolation; **B**—liver MSCs at 11 passages. Bar scales: 25 µm.

**Figure 2 cells-08-01127-f002:**
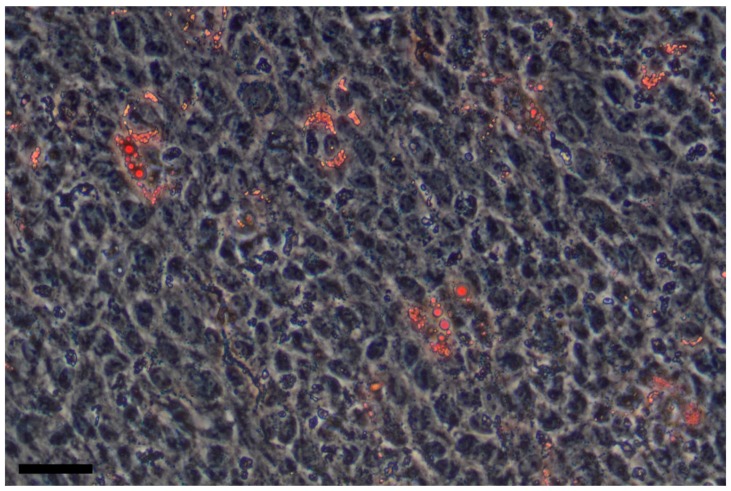
Adipogenic differentiation of liver MSCs. Bar scale: 25 µm.

**Figure 3 cells-08-01127-f003:**
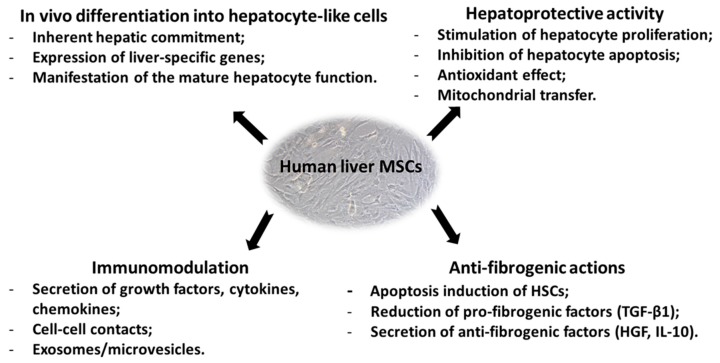
Mechanisms of therapeutic action of human liver MSCs.

**Table 1 cells-08-01127-t001:** Gene expression microarray data of human liver MSCs isolated from cirrhosis and fibrosis liver biopsies.

Probe Name	Systematic Name	Gene Name	Description	Liver2 *		Liver3 *	
**Mesenchymal Stem Cell Markers**							
A_23_P24870	NM_000610	*CD44*	Homo sapiens CD44 molecule (Indian blood group)	22,165.98	22,147.97	15,653.33	15,673.40
A_23_P36364	NM_006288	*THY1*	Homo sapiens Thy-1 cell surface antigen; CD90	1672.59	1462.77	33,192.08	28,736.66
A_23_P83328	NM_000118	*ENG*	Homo sapiens endoglin; CD105	1555.26	1482.65	2847.10	2417.84
A_23_P150053	NM_001613	*ACTA2*	Homo sapiens actin, alpha 2, smooth muscle, aorta; α-SMA	25,339.22	24,698.05	34,689.60	31,747.88
A_23_P161194	NM_003380	*VIM*	Homo sapiens vimentin	64,899.19	61,665.47	65,568.88	57,384.63
A_23_P103672	NM_006617	*NES*	Homo sapiens nestin	1438.84	1371.23	1717.24	1488.00
A_24_P119745	NM_212482	*FN1*	Homo sapiens fibronectin 1	33,852.13	31,302.16	41,280.02	37,071.62
**Pluripotent Markers**							
A_23_P204640	NM_024865	*NANOG*	Homo sapiens Nanog homeobox	2.32	2.69	2.79	3.41
A_23_P395582	NM_174900	*ZFP42*	Homo sapiens ZFP42 zinc finger protein; REX-1	2.44	2.77	2.91	3.61
A_23_P401055	NM_003106	*SOX2*	Homo sapiens SRY-box 2	6.08	8.77	8.34	3.44
A_23_P59138	NM_002701	*POU5F1*	Homo sapiens POU class 5 homeobox 1; Oct3/4	313.82	315.20	36.68	40.28
**Integrins**							
A_23_P256334	NM_181501	*ITGA1*	Homo sapiens integrin subunit alpha 1; CD49a or VLA-1	2206.52	2279.19	1514.58	1342.72
A_32_P208076	NM_002203	*ITGA2*	Homo sapiens integrin subunit alpha 2; CD49b or VLA-2	2257.31	2115.44	2226.08	2029.36
A_23_P55251	NM_002204	*ITGA3*	Homo sapiens integrin subunit alpha 3; CD49c or VLA-3	1045.91	974.42	2288.12	2026.87
A_23_P56505	NM_000885	*ITGA4*	Homo sapiens integrin subunit alpha 4; CD49d or VLA-4	628.68	628.96	709.31	667.34
A_23_P36562	NM_002205	*ITGA5*	Homo sapiens integrin subunit alpha 5; CD49e or VLA-5	1206.74	1296.71	1207.25	1142.43
A_23_P210176	NM_000210	*ITGA6*	Homo sapiens integrin subunit alpha 6; CD49f or VLA-6	526.16	587.86	908.29	907.40
A_23_P206022	NM_001004439	*ITGA11*	Homo sapiens integrin subunit alpha 11	1745.93	1722.10	8940.13	8343.51
A_23_P50907	NM_002210	*ITGAV*	Homo sapiens integrin subunit alpha-V; CD51	6462.36	6587.11	7948.93	7574.29
A_23_P128084	NM_002206	*ITGA7*	Homo sapiens integrin subunit alpha 7	1268.72	1201.85	35,025.66	32,855.67
A_23_P218375	NM_002208	*ITGAE*	Homo sapiens integrin subunit alpha E; CD103	3050.15	2959.34	2299.69	2459.11
**Cadherins**							
A_23_P38732	NM_001792	*CDH2*	Homo sapiens cadherin 2	3149.20	3461.18	2933.76	2428.21
A_23_P17593	NM_001794	*CDH4*	Homo sapiens cadherin 4	906.85	865.44	294.76	261.65
A_23_P214011	NM_004932	*CDH6*	Homo sapiens cadherin 6	262.23	253.28	127.30	120.94
A_23_P152305	NM_001797	*CDH11*	Homo sapiens cadherin 11	6958.79	6745.47	6684.07	6513.42
A_23_P40192	NM_021248	*CDH22*	Homo sapiens cadherin 22	464.46	385.37	409.64	360.69
A_23_P25790	NM_022478	*CDH24*	Homo sapiens cadherin 24	5737.23	3785.29	5150.02	2650.23
**Liver-Specific Genes**							
A_23_P205531	NM_001282192	*RNASE4*	Homo sapiens ribonuclease A family member 4	1136.06	1122.61	1057.69	931.28
A_23_P127584	NM_006169	*NNMT*	Homo sapiens nicotinamide N-methyltransferase	29,236.59	29,396.42	37,759.10	35,145.76
A_24_P53976	NM_002065	*GLUL*	Homo sapiens glutamate-ammonia ligase	852.81	794.77	543.40	525.56
A_23_P120809	NM_001288833	*GGT1*	Homo sapiens gamma-glutamyltransferase 1	687.35	622.17	2210.95	1819.73
A_23_P209625	NM_000104	*CYP1B1*	Homo sapiens cytochrome P450 family 1 subfamily B member 1	4169.50	4678.15	3127.02	3027.10
A_23_P206110	NM_000761	*CYP1A2*	Homo sapiens cytochrome P450 family 1 subfamily A member 2	1589.79	1231.16	1202.83	979.58
A_23_P257834	NM_000477	*ALB*	Homo sapiens albumin	2.29	2.69	2.71	4.04
A_23_P58205	NM_001134	*AFP*	Homo sapiens alpha fetoprotein	2.52	2.69	2.85	3.60
A_23_P28761	NM_001030004	*HNF4A*	Homo sapiens hepatocyte nuclear factor 4 alpha	4.49	4.69	3.15	3.91
A_23_P359245	NM_000245	*MET*	Homo sapiens MET proto-oncogene, receptor tyrosine kinase	8524.08	8927.24	4795.02	4726.29
**Immune Markers**							
A_23_P201758	NM_002389	*CD46*	Homo sapiens CD46 molecule	1161.59	1128.90	1383.52	1274.95
A_23_P374862	NM_000574	*CD55*	Homo sapiens CD55 molecule (Cromer blood group)	1017.57	1029.88	513.73	507.53
A_24_P141481	NM_203330	*CD59*	Homo sapiens CD59 molecule (CD59 blood group)	36,012.01	40,734.98	49,512.49	41,315.28
A_23_P208293	NM_001042724	*NECTIN2*	Homo sapiens nectin cell adhesion molecule 2; CD112	9739.88	10,486.87	10,514.83	9845.87
A_23_P121480	NM_001004196	*CD200*	Homo sapiens CD200 molecule	50.63	48.11	73.49	67.06
A_23_P256487	NM_014143	*CD274*	Homo sapiens CD274 molecule; PD-L1	2114.17	2051.64	755.44	723.45
**Antioxidant Enzymes**							
A_23_P154840	NM_000454	*SOD1*	Homo sapiens superoxide dismutase 1	51,374.15	51,358.13	55,628.91	52,328.87
A_23_P254741	NM_003102	*SOD3*	Homo sapiens superoxide dismutase 3	621.46	582.76	2163.03	2139.94
A_23_P202658	NM_000852	*GSTP1*	Homo sapiens glutathione S-transferase pi 1	68,476.57	71,676.05	62,832.28	59,208.49

* **Liver2**: Female, 35 years old, liver fibrosis (F2). **Liver3**: Male, 48 years old, liver cirrhosis of mixed (HCV and alcoholic) etiology, compensated, class A (Child–Puqh score). Liver MSCs were obtained as described (ref. [33]). Liver MSCs at passage 6 were used for analysis. The data were obtained with The Human Gene Expression Microarray Kit (Agilent; USA).

**Table 2 cells-08-01127-t002:** Phenotype of liver MSCs.

**Mesenchymal Markers**		
CD13	Membrane alanyl aminopeptidase	+ [[34],[36],]
CD44	Cell adhesion molecule; receptor for hyaluronic acid	+ [20,21,22,23,33,36,71]
CD73	Ecto-5’-nucleotidase	+ [20,21,22,34,36,71]
CD90	Thymocyte antigen (Thy-1)	+ [20,21,22,23,33,34,36,71]
CD105	Endoglin	+ or ± [21,22,30,34,36,71] − [20]
CD146	Melanoma cell adhesion molecule (MCAM)	+ [71] ± [36]
CD166	Activated leukocyte cell adhesion molecule (ALCAM)	+ [22,36]
Vimentin	Type III intermediate filament	+ [20]
α-SMA	Alpha-smooth muscle actin	+ [20]
Fibronectin	Glycoprotein of the extracellular matrix	+ [34,37]
**Hematopoietic/Endothelial Markers**		
CD11b	Integrin alpha M (ITGAM)	− [20,23,27,33,36]
CD14	Myeloid cell marker; co-receptor for lipopolysaccharide	− [20,23,27,33,36]
CD19	B-lymphocyte antigen	− [20,23,27,33,36]
CD31	Platelet endothelial cell adhesion molecule (PECAM-1)	− [20,23,27,33,36]
CD34	Hematopoietic stem cell marker	− [20,23,27,33,36]
CD45	Leukocyte common antigen	− [20,23,27,33,36]
CD79β	B-lymphocyte antigen	− [20,23,27,33,36]
CD117	Mast/stem cell growth factor receptor (c-Kit)	− [20,23,27,33,36]
CD133	Prominin-1; marker of hematopoietic stem cells, endothelial progenitor cells, cancer stem cells etc.	− [20,23,27,33,36]
CD144	Vascular endothelial cadherin or cadherin-5	− [20,23,27,33,36]
HLA-ABC	Major histocompatibility complex class I	+ [20,23,27,33,36]
HLA-DR	Major histocompatibility complex class II	− [20,23,27,33,36]
**Pluripotency Markers**		
SSEA-3	Stage-specific embryonic antigen 3	− [36]
SSEA-4	Stage-specific embryonic antigen 4	− [36] + [21]
Tra1-60	Pluripotent stem cell markers	− [36]
Tra1-81	Pluripotent stem cell markers	− [36]
NANOG	Transcription factor that maintain pluripotency	+ [21,34]
OCT-4	Homeodomain transcription factor of the POU family; involved in the self-renewal	+ [21,34]
SOX2	Transcription factor; essential for maintaining self-renewal and pluripotency	+ [21]
REX1	Pluripotency marker	+ [34]
**Integrins and Adhesion Molecules**		
CD29	Integrin beta-1 (ITGB1)	+ [6,20,21,22,23,28,36,71]
CD49a	Integrin alpha-1 (ITGA1)	+ [34] ± [[36],]
CD49b	Integrin alpha-2 (ITGA2)	+ [20,28,34,36]
CD49c	Integrin alpha-3 (ITGA3)	+ [28,34,36]
CD49d	Integrin alpha-4 (ITGA4)	− [36]
CD49e	Integrin alpha-5 (ITGA5)	+ [20,28,34,36]
CD49f	Integrin alpha-6 (ITGA6)	± [20] − [36]
CD51	Integrin alpha-V (ITGAV)	+ [36]
CD162	Selectin P ligand (SELPLG)	− [36]
SSEA-1	Sialyl LewisX (CD15s)	− [36]
Cadherin-11	Integral membrane proteins that mediate calcium-dependent cell-cell adhesion	+ [34]
**Cytokeratins and Hepatic Markers**		
Albumin	Marker of mature hepatocytes	+ [20,21] − [22,29]
α-fetoprotein	Marker of immature hepatocytes; highly expressed in the fetal liver; can be a sign of liver cancer, as well as noncancerous liver diseases	± [20,21] − [29]
Hepatocyte nuclear factor-4 (HNF-4)	Transcription factor that plays a critical role in the transcriptional regulation of genes involved in glucose metabolism in hepatocytes	+ [20] − [29]
CYP3A4	Cytochrome P450 3A4; it oxidizes small foreign organic molecules (xenobiotics)	+ [20]
CYP1B1	Cytochrome P450 1B1; it catalyzes estrogen hydroxylation and activates potential carcinogens	+ [20]
Cytokeratin 7	Type II cytoskeletal keratin; found on many glandular and transitional epithelia, hepatocytes, biliary epithelium etc.	− [20]
Cytokeratin 8	Type II cytoskeletal keratin; expressed mainly by secretory epithelia	− [20] + [21]
Cytokeratin 18	Type I cytoskeletal keratin; expressed in single layer epithelial tissues	− [20] ± [22,72] + [21]
Cytokeratin 19	Type I cytoskeletal keratin; markers of cells of the epithelial origin	− [20,21] + [22]
c-Met	Hepatocyte growth factor receptor (HGFR)	± [22] + [29]
**Neuronal Markers**		
GFAP	Glial fibrillary acidic protein; an intermediate filament	− [37,83]
CD56	Neural cell adhesion molecule (NCAM)	− [37,83]
NT-3	Neurotrophin-3; neurotrophic factor in the nerve growth factor family	− [37,83]
CD271	Low-affinity nerve growth factor receptor (NGFR)	− [36,37,83]
Nestin	Type VI intermediate filament protein; neuroectodermal stem cell marker	+ [28]

“+”—high expression; “±”—middle or low expression; “−”—lack of expression.

**Table 3 cells-08-01127-t003:** Protocols of hepatogenic differentiation of liver MSCs.

References	Protocols	Results
[20]	1. Iscove’s Modified Dulbecco’s Medium (IMDM) containing 20 ng/mL epidermal growth factor (EGF), 10 ng/mL basic fibroblast growth factor (bFGF) for 2 days. 2. IMDM containing 20 ng/mL hepatocyte growth factor (HGF), 10 ng/mL bFGF, nicotinamide 0.61 g/L, and 1% insulin-transferrin-selenium (ITS) premix for 10 days. 3. IMDM containing 20 ng/mL oncostatin M, 1 μM dexamethasone, and 1% ITS premix for 10 days. For each step, medium was changed every 3 days.	Morphology change from elongated to polygonal shape with granular cytoplasm; expression of liver-associated genes (*TDO*, *TAP*, *CYP3A4*, *CYP2B6*); storage of glycogen.
[21]	Liver MSCs were incubated under the condition of microgravity or in flasks coated with Matrigel in α-MEM/EBM 3:1, 12 mM Hepes, 2% FCS with hepatocyte growth factor (HGF) (10 ng/mL), and fibroblast growth factor 4 (FGF4) (10 ng/mL) for 15 days.	Change in the morphology from elongated to cuboid cells; positivity for cytochrome P450 activity; synthesis and release of albumin and urea; reduction of α-fetoprotein expression; increase in cytokeratin 8 and cytokeratin 18 expression.
[72]	Preinduction medium, consisting of IMDM supplemented with 2% FBS, 20 ng/mL EGF, and 10 ng/mL FGF-4 for 2 days before induction by a 2-step protocol (protocol A). Protocol A: Step-1 differentiation medium consisting of IMDM supplemented with 2% FBS, 10 ng/mL FGF-4, 20 ng/mL HGF, and 5 mol/L nicotinamide for 7 days. Step-2 maturation medium consisting of IMDM supplemented with 2% FBS, 10 ng/mL FGF-4, 20 ng/mL oncostatin M, 1 mol/L dexamethasone, 50 g/mL ITS premix, and 1 mol/L trichostatin A. Protocol B: Step-1 differentiation medium consisting of IMDM supplemented with 2% FBS, 10 ng/mL FGF-4, 20 ng/mL HGF, 5 mol/L nicotinamide, and 2.5 mmol/L sodium butyrate for 7 days. Step-2 maturation medium consisting of IMDM supplemented with 2% FBS, 10 ng/mL FGF-4, 20 ng/mL oncostatin M, 1 mol/L dexamethasone, 50 g/mL ITS premix, 1 mol/L trichostatin A, 2.5 mmol/L sodium butyrate, and 20 ng/mL HGF. Medium changes were performed twice weekly up to 21 days in both protocols.	Expression of mature hepatocyte markers, such as albumin, cytokeratin 18, and tryptophan 2,3-dioxygenase; albumin secretion.
[27]	Liver MSCs were seeded into fibronectin-coated plates. 1. DMEM (low glucose) containing 1% ITS premix, 10 ng/mL FGF-1, 10 ng/mL FGF-4, and 20 ng/mL HGF for 5 days. 2. DMEM (low glucose) containing 100 nM dexamethasone, 10 ng/mL FGF-4, 20 ng/mL HGF, 10 ng/mL oncostatin M, and 0.5% dimethyl sulfoxide on designated days. Medium was exchanged every 2 days.	The morphology was changed to a highly round or polygonal shape; expression of some hepatic markers, such as albumin, α1-antitrypsin, tryptophan 2,3-dioxygenase, and glutamine synthetase; urea production; CYP450 enzymatic functions were not detected.
[34]	Liver MSCs were seeded into fibronectin-coated dishes at high density and maintained for two weeks in a medium containing 0.5% FBS, 10 ng/mL FGF-4, and 20 ng/mL HGF. After this period, FGF-4 and HGF were substituted for 20 ng/mL oncostatin M for other 14 days.	The cell morphology was changed to a globular shape with an eccentric nucleus; expression of cytokeratins 18 and 19 and transcription factor GATA-4; glycogen storage, and albumin production; inducible cytochrome P450 activity.
[33]	1. DMEM/F12 with 2 mM GlutaMAX supplemented with 0.5% FBS, 10 ng/mL HGF, and 100 ng/mL Activin A. 2. DMEM/F12 medium with 2 mM GlutaMAX containing 15 mM HEPES, 2 μg/mL insulin, 10 ng/mL FGF-4, 10 ng/mL HGF, 10 ng/mL oncostatin M and 10^−7^ M dexamethasone. In each medium, cells were cultivated for 10 days with medium change every 2 days.	The cell morphology was more flattened with a cuboidal epithelium-like shape and granular cytoplasm; prealbumin expression; reduction of α-fetoprotein expression; albumin secretion.

**Table 4 cells-08-01127-t004:** Clinical trials of human liver MSCs for patients with liver disorders.

Status	Clinical Trials.gov Identifier	Study Title	Conditions	Interventions	Phase
Recruiting	NCT03884959	A Prospective, Open-Label, Safety, and Efficacy Study of Infusions of HepaStem in Urea Cycle Disorders Pediatric Patients	Urea Cycle Disorder	HepaStem	II
Recruiting	NCT02946554	Multicenter Phase II Safety and Preliminary Efficacy Study of 2 Dose Regimens of HepaStem in Patients with Acute-on-Chronic Liver Failure	Acute-On-Chronic Liver Failure	HepaStem	II
Recruiting	NCT03963921	Multicenter, Open-Label, Safety and Tolerability Study of Ascending Doses of HepaStem in Patients with Cirrhotic and Pre-cirrhotic Non-alcoholic Steato-hepatitis (NASH)	Non-Alcoholic Steatohepatitis	HepaStem	I/II
Enrolling by invitation	NCT03343756	HepaStem Long-Term Safety Registry (PROLONGSTEM)	Urea Cycle Disorder Crigler–Najjar Syndrome	HepaStem	
Enrolling by invitation	NCT03632148	In Vitro Evaluation of the Effect of HepaStem in the Coagulation Activity in Blood of Patients with Liver Disease	Decompensated Cirrhosis	HepaStem	
Completed	NCT01765283	A Prospective, Open-Label, Multicenter, Partially Randomized, Safety Study of One Cycle of Promethera HepaStem in Urea Cycle Disorders (UCD) and Crigler–Najjar Syndrome (CN) Pediatric Patients	Urea Cycle Disorders, Crigler–Najjar Syndrome	HepaStem	I/II
Completed	NCT02051049	Long-term Safety Follow-up Study of Patients With Received Infusions of HepaStem	Urea Cycle Disorders Crigler–Najjar Syndrome	HepaStem	
Unknown	NCT02489292	Prospective, Open-Label, Multicenter, Efficacy, and Safety Study of Several Infusions of HepaStem in Urea Cycle Disorders Pediatric Patients	Urea Cycle Disorders	HepaStem	II

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
