# Peer review of "Mesenchymal Stem Cells in the Adult Human Liver: Hype or Hope?"

_cells, 2019, doi:10.3390/cells8101127_

Round 1

Reviewer 1 Report

This is an interesting well-described study, , well written, that explores Mesenchymal stem cells from liver. It addresses an interesting clinical area in an application. The focus of the manuscript is within the scope of the journal and the guidelines of the targeted journal are followed. English grammar, style, and syntax are correct. Congratulations to the authors for this excellent review

Minor revisions:

What weakens this review is:

1) A detailed comparison of the different sources of MSC according to the tissue from which they are collected, with a comparative table.

2) It is necessary to demonstrate the value of using liver MSCs rather than fat tissue that is easier to isolate.

3) A summary scheme of the mode of action of liver MSC should be added. A separate paragraph should be devoted to biodistribution and another to animal models. A scale or magnification must be indicated for the photos. Line 365, the three differentiation pathways must be treated; i. e. osteo, chondro and adipocyte; with photos corresponding to the 3 differentiation pathways.

4) In order to highlight clinical trials, it is necessary to devote a paragraph to them with a complete table that will be placed at the end of the article.

Reviewer 2 Report

 see attached 

Round 2

Reviewer 2 Report

it addressed my concerns . I have no more comments .